# A hole-selective hybrid TiO₂ layer for stable and low-cost photoanodes in solar water oxidation

Sanghyun Bae[1,2], Thomas Moehl[1], Erin Service[1], Minjung Kim[2], Pardis Adams[1], Zhenbin Wang[1], Yuri Choi[2], Jungki Ryu[2,3] ✉ & S. David Tilley[1] ✉

The use of conductive and corrosion-resistant protective layers represents a key strategy for improving the durability of light absorber materials in photoelectrochemical water splitting. For high performance photoanodes such as Si, GaAs, and GaP, amorphous TiO₂ protective overlayers, deposited by atomic layer deposition, are conductive for holes via a defect band in the TiO₂. However, when coated on simply prepared, low-cost photoanodes such as metal oxides, no charge transfer is observed through amorphous TiO₂. Here, we report a hybrid polyethyleneimine/TiO₂ layer that facilitates hole transfer from model oxides BiVO₄ and Fe₂O₃, enabling access to a broader scope of available materials for practical water oxidation. A thin polyethyleneimine layer between the light absorber and the hybrid polyethyleneimine/TiO₂ acts as a hole-selective interface, improving the optoelectronic properties of the photoanode devices. These polyethyleneimine/TiO₂ modified photoanodes exhibit high photostability for solar water oxidation over 400 h.

Photoelectrochemical (PEC) water splitting is a promising route to low-cost and large-scale green hydrogen production. The durability of the light absorbers immersed in the aqueous electrolyte solution is a major factor in the cost of the resulting hydrogen[1,2], and much of the efforts on simple semiconductor/electrolyte junction PEC systems have shifted towards incorporating protective overlayers to minimize (photo) corrosion of the materials. The use of such overlayers aids not only the durability, but also the optoelectronic properties of the absorber layer: the photovoltage in these coated systems is generated at a solid-solid interface, and various strategies such as contact selectivity can be used to improve the photovoltage[3,4].

Amorphous titanium dioxide (a-TiO₂) deposited by atomic layer depositions (ALD) has been widely employed for corrosion protection of PEC materials, primarily for photocathodes, since the conduction band of a-TiO₂ is close in energy to the thermodynamic hydrogen evolution potential[5,6]. For smaller bandgap, high-efficiency photoanode materials, the deep valence band of the a-TiO₂ overlayer represents a large barrier for hole injection that could preclude charge transfer. However, a defect band in the a-TiO₂ deposited by ALD was shown to enable hole transfer, and photoanode materials such as crystalline Si, GaAs, and GaP modified with a-TiO₂ have demonstrated excellent performance and stability for PEC water oxidation[7].

Despite these demonstrations of photoanode stabilization using energy-intensive and/or high-cost semiconductors, the use of a-TiO₂ layers on low-cost, easy-to-prepare photoanode materials, such as metal oxides[8], has been rarely reported, likely due to high recombination at the photoanode/TiO₂ interface from the relatively slow extraction of holes through the a-TiO₂ defect band[9,10]. While metal oxides are investigated for large-scale water splitting due to their Earth abundance, ease of synthesis and perceived stability, they often suffer from (photo)corrosion[11,12], and therefore efforts have been directed towards stabilizing them. For instance, McDowell et al. used a-TiO₂ as a protective layer for BiVO₄ photoanodes that showed stability for several hours[13]. However, the thickness of the protective layer was only 1 nm, and charge transfer could be achieved via tunneling. For long-term stability, there should ideally be no pinholes in the protective layer,

[1]Department of Chemistry, University of Zurich, Winterthurerstrasse 190, 8057 Zurich, Switzerland. [2]School of Energy and Chemical Engineering, Ulsan National Institute of Science and Technology (UNIST), Ulsan 44919, Republic of Korea. [3]Center for Renewable Carbon, Ulsan National Institute of Science and Technology (UNIST), Ulsan 44919, Republic of Korea. ✉e-mail: jryu@unist.ac.kr; david.tilley@chem.uzh.ch

and to have a good chance of achieving this, relatively thick layers of TiO₂ will likely be required (> 50 nm)[14].

In this context, polyethyleneimine (PEI), which consists of repeating units of amine groups and two-carbon aliphatic spacers, was selected to alleviate unfavorable interface energetics and facilitate the hole-selective transfer. The non-conjugated PEI is known not only as a modifier of work function[15,16] but also as a hole transfer channel[17,18] since the amine moieties are easily oxidized, enabling hole transfer in PEC devices. We considered these unique properties of PEI to be promising for effectively addressing the aforementioned issues at the interface between metal oxides and a-TiO₂.

Here, we report a hybrid PEI/TiO₂ layer on both BiVO₄ and Fe₂O₃ that not only protects these relatively small bandgap metal oxide photoanode materials from (photo)corrosion but also serves as a hole-selective contact. Although a PEI coating has been reported to lower the work function of semiconductor materials[15], in this study, we embed the PEI in a-TiO₂ during the ALD process by reacting the TiO₂ precursor with the abundant amine functionalities of the polymer, yielding a highly defective PEI/TiO₂ layer that transmits holes and blocks electrons. The BiVO₄/PEI/TiO₂ photoanode demonstrated an onset potential of 0.28 V vs. reversible hydrogen electrode (RHE), a photocurrent of 2.03 mA cm⁻² at 1.23 vs. RHE, and stable PEC water oxidation for 400 h in pH 8 electrolyte solution.

## Results and Discussion

### Preparation of photoanodes and evaluation of PEC water oxidation efficiency

We deposited a PEI layer onto the BiVO₄ surface by spin-coating with a 4 wt.% aqueous PEI solution, followed by the deposition of nominally 100 nm a-TiO₂ by ALD (BiVO₄/PEI/TiO₂) (Fig. 1). For comparison, we also prepared a photoanode without the PEI layer using the same procedure (BiVO₄/TiO₂). First, we compared the surface morphology of BiVO₄/TiO₂ and BiVO₄/PEI/TiO₂ by scanning electron microscopy (SEM). According to SEM measurements, the TiO₂ layer was conformally deposited on the highly porous BiVO₄ photoanodes, regardless of the presence of a PEI interfacial layer (Supplementary Fig. 1). The conformal and uniform layer of a-TiO₂ was further confirmed by a pinhole test using cyclic voltammetry in a ferricyanide solution (Supplementary Figs. 2 and 3)[14]. The morphology of BiVO₄/TiO₂ and BiVO₄/PEI/TiO₂ photoanodes was slightly different. BiVO₄/PEI/TiO₂ exhibited lower porosity in comparison to BiVO₄/TiO₂. This result could be attributed to the PEI layer filling the pores within the porous BiVO₄ structure (Supplementary Fig. 4). The thickness of the PEI layer on the BiVO₄ surface was estimated as 50.5 nm through profilometer measurement on an FTO electrode (Supplementary Fig. 5).

After confirming the uniform deposition of the TiO₂ protection layer, we evaluated the PEC performance and the stability of the photoanodes. Linear sweep voltammetry (LSV) was carried out in 0.5 M potassium phosphate (KPi) buffer solution (pH 8). First, the BiVO₄/TiO₂ photoanode exhibited a very low photocurrent density of 0.0008 mA cm⁻² at 1.23 V vs. RHE under 1 sun illumination (Fig. 2a). Even after the modification with CoOOH co-catalyst (BiVO₄/TiO₂/CoOOH), the photocurrent density only remained in the microampere (μA cm⁻²) range at 1.23 V vs. RHE. The ALD TiO₂ used here has been previously shown to transmit holes on silicon photoanodes[19], and thus the typical leaky TiO₂ is not suitable for BiVO₄.

Previous studies have established that photoanodes modified with a-TiO₂ and metal co-catalysts exhibited enhanced PEC performance, attributed to the sufficient charge extraction from the TiO₂ to the co-catalysts[20,21]. However, even with various types of metal co-catalysts, we observed insufficient improvement and non-reproducibility in the PEC performance according to the work function of the used metal (Supplementary Fig. 6). This result indicates that the photogenerated holes are unable to be transferred through the amorphous TiO₂ layer when the thick TiO₂ is directly deposited on BiVO₄. On the contrary, BiVO₄/PEI/TiO₂ showed photoanodic current even without co-catalysts: photocurrent density of 1.08 mA cm⁻² at 1.23 V vs. RHE (Fig. 2b). Since co-catalysts are generally required for water oxidation[7], the photoanodic current may (partly) originate from the oxidation of the PEI layer at the interface between BiVO₄ and TiO₂ layer[22,23]. Further details on the origin of the photocurrent in BiVO₄/PEI/TiO₂ will be discussed in a subsequent section.

We also prepared co-catalyst-modified photoanodes (BiVO₄/CoOOH and BiVO₄/PEI/TiO₂/CoOOH) to facilitate efficient water oxidation (Supplementary Fig. 7 and 8). The CoOOH was selected due to its superior catalytic activity compared to FeOOH and NiOOH deposited via the immersion process (Supplementary Fig. 9). The BiVO₄/PEI/TiO₂/CoOOH photoanode exhibited a notable increase in photocurrent of 2.03 mA cm⁻² (at 1.23 V vs. RHE), along with an onset potential of 0.28 V vs. RHE, estimated by extrapolating the linear rising region of the photocurrent to the x-axis (Fig. 2c). The applied bias photon-to-current efficiency (ABPE) showed maximum efficiencies of 0.15%, 0.61%, and 0.73% for the BiVO₄, BiVO₄/CoOOH, and BiVO₄/PEI/TiO₂/CoOOH, respectively (Supplementary Fig. 10). We also evaluated the photoconversion efficiency of BiVO₄, BiVO₄/PEI/TiO₂, and BiVO₄/PEI/TiO₂/CoOOH electrodes through the incident photon-to-current efficiency (IPCE) measurement (Supplementary Fig. 11 and 12). In the measurement, the photoanodes with the overlayers exhibited much higher conversion efficiencies at longer wavelengths near the band gap.

Subsequently, we evaluated the stability of the photoanodes by chronoamperometry (CA) in PEC water oxidation. While the bare BiVO₄ and BiVO₄/CoOOH photoanodes rapidly degraded within 5 h, the BiVO₄/PEI/TiO₂/CoOOH maintained its PEC activity for an

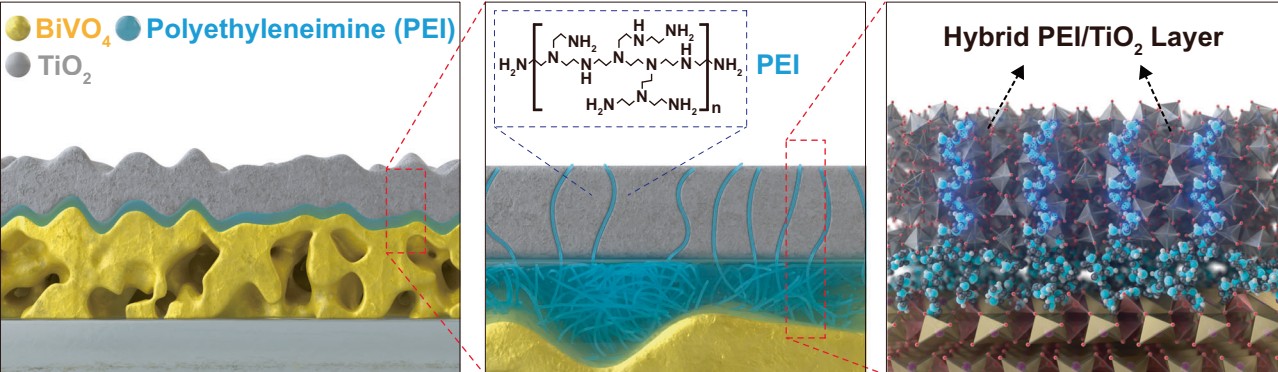

**Fig. 1 | Configuration of interfacial PEI and hybrid PEI/TiO₂ layer.** Graphical illustration of a modified BiVO₄ photoanode featuring a thin, insulating PEI layer between the BiVO₄ and a hole-conductive hybrid PEI/TiO₂ layer.

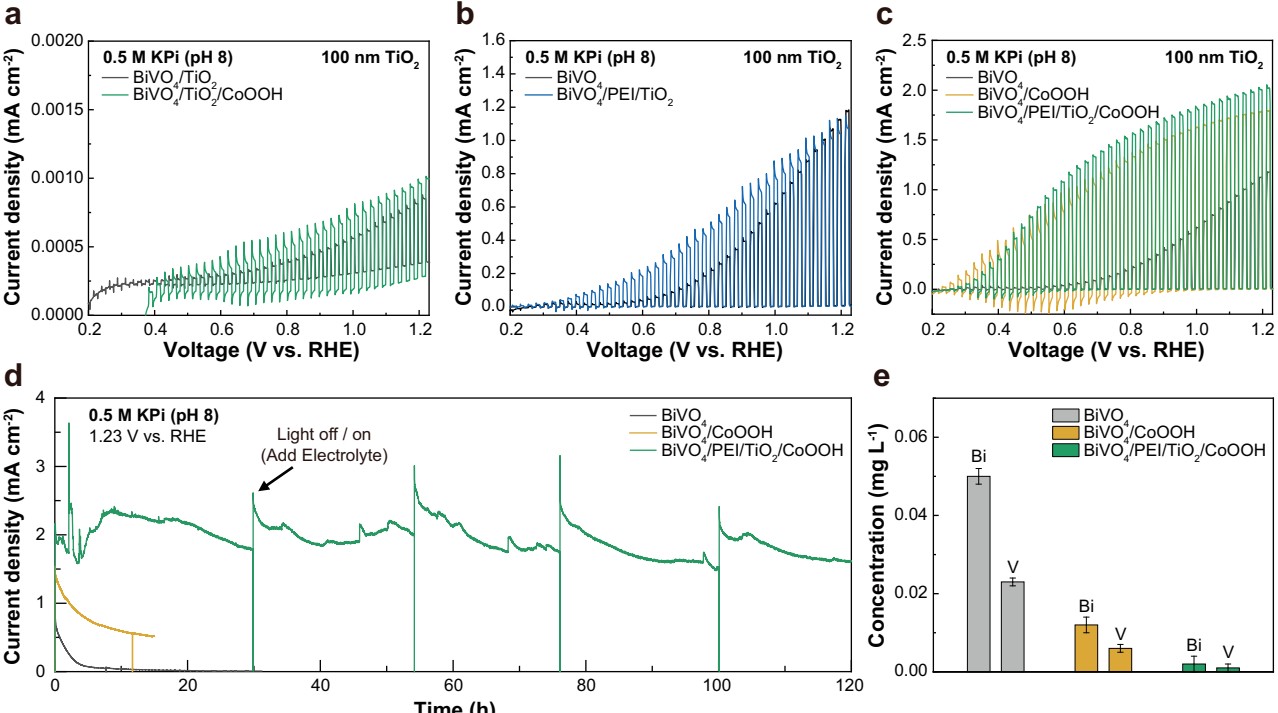

**Fig. 2 | Evaluation of PEC water oxidation of BiVO₄ photoanode modified with PEI and amorphous TiO₂. a–c** LSV curves of BiVO₄/TiO₂ with and without CoOOH (**a**), bare BiVO₄ and BiVO₄/PEI/TiO₂ (**b**), and BiVO₄/CoOOH and BiVO₄/PEI/TiO₂/ CoOOH **c. e** Concentration of dissolved metal ion from each photoanode during stability test. The error bars represent the standard deviations of triplicate experiments.

extended duration over 120 h, achieving 96% Faradaic efficiency for oxygen evolution as measured by gas chromatography (GC) (Fig. 2d and Supplementary Figs. 13 and 14). The SEM measurement revealed negligible changes in morphology compared to the photoanode before the stability test (Supplementary Fig. 15), and high-resolution XPS spectra showed a clear Ti 2p peak in BiVO₄/PEI/TiO₂/CoOOH even after the 120 h stability test (Supplementary Fig. 16). The stability of BiVO₄/PEI/TiO₂/CoOOH was also enhanced even under alkaline conditions (pH 11), in which BiVO₄ is inherently unstable (Supplementary Fig. 17). Moreover, inductively coupled plasma-optical emission spectroscopy (ICP-OES) measurement demonstrated a negligible dissolution of Bi and V from BiVO₄ to the electrolyte after the stability test of BiVO₄/PEI/TiO₂/CoOOH (Fig. 2e). The PEI/TiO₂ layer therefore offers corrosion protection in addition to conductivity for holes (Table S1).

## Characterization of the BiVO₄ photoanodes

Next, we determined the elemental composition of the photoanodes before and after deposition of TiO₂ and PEI layer by X-ray photoelectron spectroscopy (XPS) analysis (Fig. 3a). The peaks of Bi (4$f$, 4$d$, and 4$p$), V (2$p$), and O (1$s$) were observed in the bare BiVO₄. The Bi and V peaks disappeared upon TiO₂ deposition, and additional peaks of Ti (2$p$), N (1$s$), and C (1$s$) appeared, commonly originating from the TiO₂ precursor tetrakis(dimethyl-amido)titanium (TDMAT)[7]. The disappearance of the Bi, V, and O peaks is attributed to the formation of a uniform and thick TiO₂ protective layer on top of the BiVO₄ surface. BiVO₄/PEI/TiO₂ also indicated a similar trend with BiVO₄/TiO₂ but showed negatively shifted C peaks at 285.6 eV and 287.4 eV, corresponding to C-NH₂ and C-NHR, respectively, along with stronger N peaks (Supplementary Figs. 18 and 19)[24,25]. Especially, the high intensity of the N peak is presumed to originate from PEI and suggests the incorporation of PEI within the amorphous TiO₂ layer.

To further investigate the composition and structure of TiO₂, we carried out time-of-flight secondary ion mass spectrometry (TOF-

SIMS) and X-ray diffraction (XRD) analyses. In the TOF-SIMS measurement, the elemental composition of C, N, and Ti within the TiO₂ layer on both BiVO₄/TiO₂ and BiVO₄/PEI/TiO₂ photoelectrodes was confirmed. Although all elements are present in both samples, the C and N signals are much higher in the BiVO₄/PEI/TiO₂ (Fig. 3b and Supplementary Fig. 20). Especially, BiVO₄/PEI/TiO₂ exhibited a significantly higher intensity of C and N near the TiO₂ surface (Supplementary Fig. 20a–b), which is consistent with the XPS analysis. XRD analysis exhibited negligible changes in the phases of BiVO₄ and an amorphous TiO₂ with no diffraction peaks regardless of the presence of PEI (Fig. 3c). These results reveal that the PEI interfacial layer does not affect the amorphous nature of the TiO₂ but does contribute to an increase in the amount of embedded C and N, altering the intrinsic properties of the a-TiO₂.

The cross-sectional images and configurations of the photoelectrodes were investigated using high-resolution transmission electron microscopy (HRTEM) and energy dispersive X-ray (EDX) spectroscopy to analyze the configurations of the hybrid PEI/TiO₂ layer. In the HRTEM measurement, the BiVO₄/TiO₂ photoanode showed a conformally deposited 100 nm TiO₂ layer that extended deep into the BiVO₄ pores (Fig. 4a), and EDX analysis also clearly revealed a uniform distribution of Ti and O throughout the entire BiVO₄ surface (Fig. 4b–d and Supplementary Fig. 21a–b). However, we observed the formation of a 125 nm thick TiO₂ layer in BiVO₄/PEI/TiO₂ and the presence of an interfacial PEI layer with a thickness ranging from 1 to 12 nm between BiVO₄ and TiO₂ (Fig. 4e and Supplementary Fig. 21c–d), and EDX analysis revealed the absence of Ti and O within the BiVO₄ structure (Fig. 4f–h). This result indicates that the ALD precursor TDMAT could penetrate the PEI layer some tens of nanometers (to within a nanometer of the interface with BiVO₄), resulting in a thicker ALD TiO₂ layer containing embedded PEI polymer. As demonstrated in molecular layer deposition techniques, it is presumed that the TDMAT precursor chemisorbed onto the amine groups of the PEI polyelectrolyte[26,27].

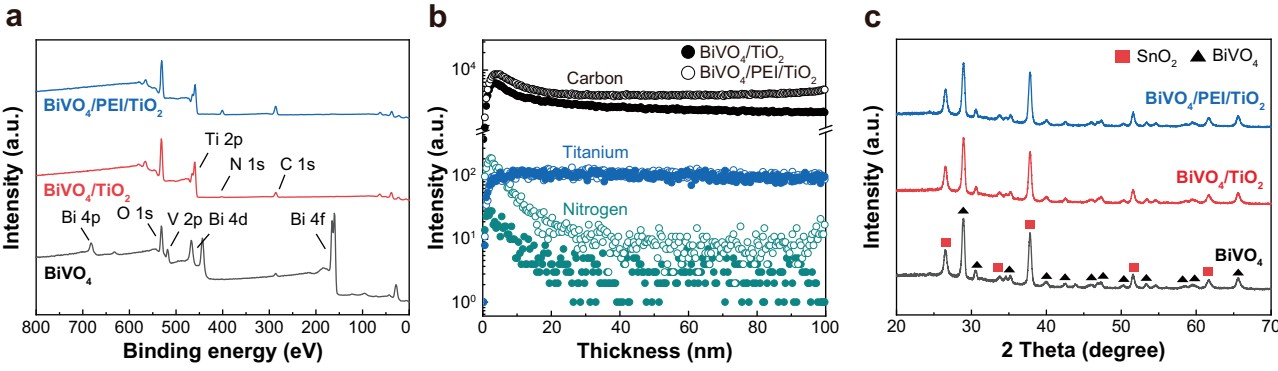

**Fig. 3 | Characterization of each BiVO₄ photoanode. a–c** XPS (**a**), TOF-SIMS (**b**), and XRD (**c**) analysis of BiVO₄, BiVO₄/TiO₂, and BiVO₄/PEI/TiO₂.

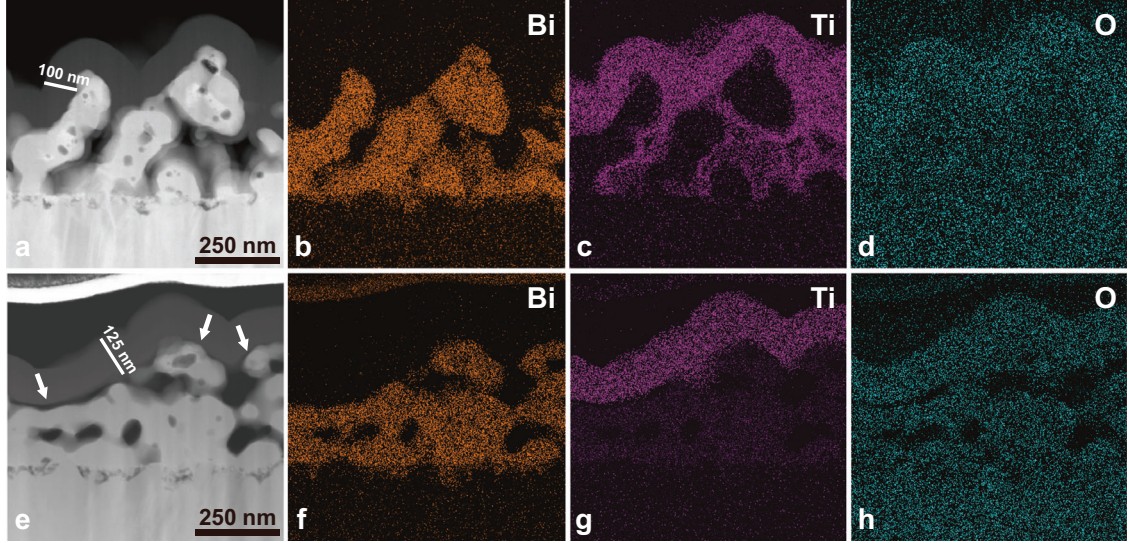

**Fig. 4 | Cross-sectional HRTEM image and EDX analysis of BiVO₄/TiO₂ and BiVO₄/PEI/TiO₂. a–d** Cross-sectional image of BiVO₄/TiO₂ (**a**) and elemental mapping of Bi, Ti, and O (**b–d**), respectively. **e–h** Cross-sectional image of BiVO₄/ PEI/TiO₂ (**e**) and the corresponding mapping elements (**f–h**). White arrows indicate the interfacial PEI layer between BiVO₄ and TiO₂.

We next investigated the PEC performance of BiVO₄/PEI/TiO₂ as a function of the thickness of the PEI layer. We prepared PEI layers with different thicknesses on the BiVO₄ surface by adjusting the concentration of PEI solution (1 wt.%, 2 wt.%, and 4 wt.%), followed by the deposition of a 100 nm TiO₂ layer on top of them. The thicknesses of PEI layer also estimated using a profilometer on a FTO substrate were 22.1 nm (1 wt.%), 35.5 nm (2 wt.%) (Supplementary Fig. 22), and 50.5 nm (4 wt.%) (Supplementary Fig. 5). The thickness of PEI is likely different on BiVO₄ surfaces, but it could be nevertheless tuned by adjusting the concentration of the PEI in solution. LSV measurements showed variations in PEC performance depending on the thickness of the PEI layer. The photoelectrode with a thicker PEI layer exhibited a higher photocurrent compared to the one with a thinner PEI layer (Supplementary Fig. 23). In addition, TOF-SIMS measurement indicated lower concentrations of C and N in TiO₂ with a thinner PEI layer (Supplementary Fig. 24). These results imply that the amount of embedded PEI may influence the properties of the hybrid PEI/TiO₂, thereby contributing the PEC performance. As previously mentioned, we considered the possibility that the photoanodic current of BiVO₄/PEI/TiO₂ originated from the oxidation of the thin interfacial PEI layer. However, the fact that only certain thicknesses of PEI coatings give photocurrent suggests that the hybrid PEI/TiO₂ layer has sufficient catalytic activity to facilitate water oxidation without a co-catalyst, although the stability is limited.

To confirm that the phenomenon is generalizable to other materials, we extended our investigation to Fe₂O₃, a well-studied semiconductor photoanode known for having slow hole transfer. We deposited a 100 nm a-TiO₂ layer onto an Fe₂O₃ photoanode with and without a PEI layer using the same method explained before. The XPS and SEM measurements showed the same trend compared to BiVO₄ (Supplementary Fig. 25 and 26), and the Fe₂O₃/TiO₂ also exhibited a very low photocurrent density in the microampere scale (0.0018 mA cm⁻² at 1.6 V vs. RHE) similar to that of BiVO₄/TiO₂ (Supplementary Fig. 27a–b). However, we observed improved PEC performance and the dramatic shift in onset potential with a thicker PEI interfacial layer under the TiO₂ protective layer (Supplementary Fig. 27b–c), which indicates that the thickness of PEI significantly influences PEC performance of the hybrid PEI/TiO₂ layer.

## Investigation of defect band in the hybrid PEI/TiO₂

It was assumed that the presence of PEI moiety in hybrid PEI/TiO₂ could influence the intrinsic properties of amorphous TiO₂. To elucidate the intrinsic properties of the hybrid PEI/TiO₂, we conducted electron energy-loss spectroscopy (EELS) for BiVO₄/TiO₂ and BiVO₄/PEI/TiO₂ photoanode. This analysis aimed to characterize the defect band of a-TiO₂, which is crucial for facilitating hole transfer from light absorbers to drive water oxidation. In EELS analysis, Cs-corrected TEM was used with a line scan technique for depth profiling analysis to investigate the

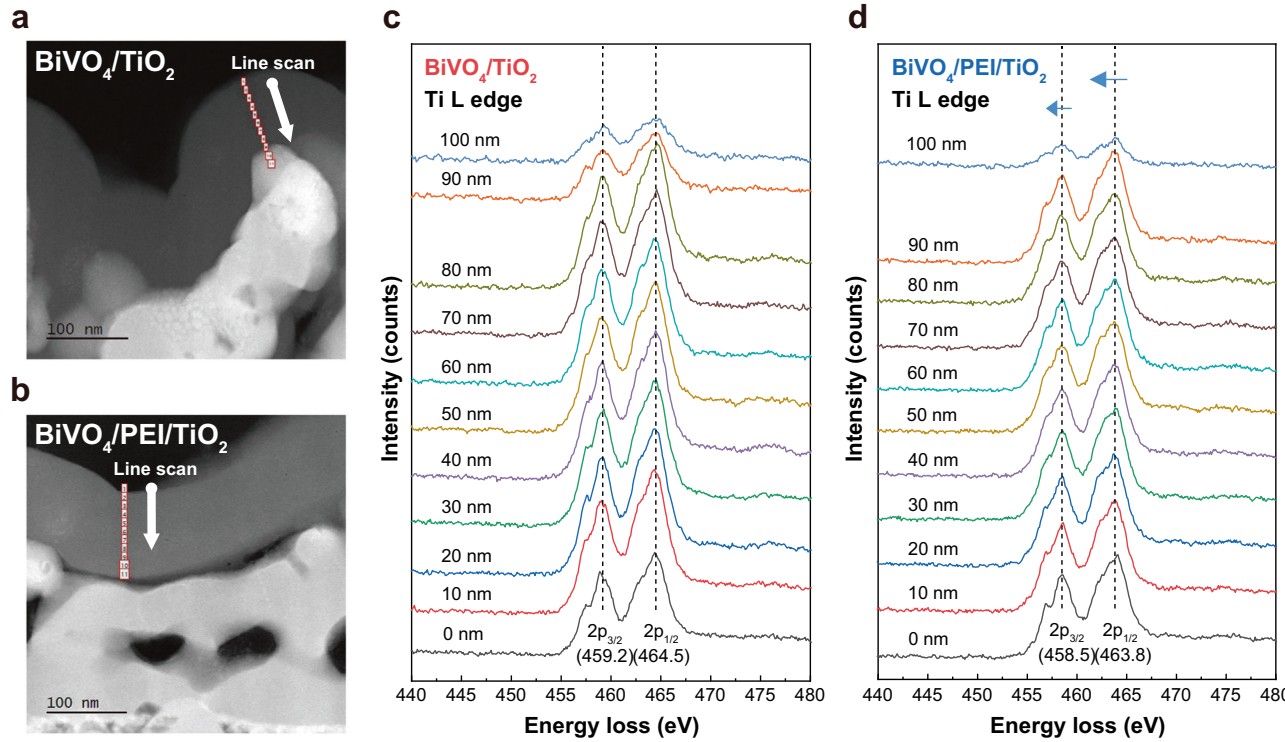

**Fig. 5 | EELS analysis for confirming oxidation state of Ti. a, b** Cs-corrected TEM images of BiVO$_4$/TiO$_2$ and BiVO$_4$/PEI/TiO$_2$ with the probing path of line scan. **c, d** The EELS spectra of the Ti L edge obtained from the certain region of BiVO$_4$/TiO$_2$ (**c**) and BiVO$_4$/PEI/TiO$_2$ (**d**).

oxidation state of Ti and O within both the amorphous TiO$_2$ and the hybrid PEI/TiO$_2$ layer (Fig. 5a, b and Supplementary Fig. 28 and 29). EELS analysis of BiVO$_4$/TiO$_2$ showed two primary peaks in the Ti L-edge (459.2 and 464.5 eV) and O K-edge spectrum (531.4 and 542.65 eV), and we verified that the position of each peak remained consistent regardless of the depth (Fig. 5c and Supplementary Fig. 30 and 31). However, in the case of BiVO$_4$/PEI/TiO$_2$, while there were no depth-dependent peak variations, we observed an overall shift of Ti peaks towards the low-energy region (Fig. 5d and Supplementary Fig. 32 and Table S2 and S3). The low-energy shift of Ti is known to occur due to the reduction of Ti$^{4+}$ in TiO$_2$, which suggests that the hybrid PEI/TiO$_2$ exhibits a higher Ti$^{3+}$ population with an excess electron compared to normal a-TiO$_2$[28,29]. The reduction of Ti was also confirmed through high resolution XPS analysis and Kelvin probe force microscopy (KPFM) measurements for both normal TiO$_2$ and hybrid PEI/TiO$_2$. The peak deconvolution of Ti 2p$_{1/2}$ and 2p$_{3/2}$ in both electrodes showed a slightly higher proportion of Ti$^{3+}$ in the hybrid PEI/TiO$_2$ (Supplementary Fig. 33)[30,31]. The KPFM measurements revealed work functions of 4.82 eV, 4.57 eV, and 4.41 eV for the BiVO$_4$, BiVO$_4$/TiO$_2$, and BiVO$_4$/PEI/TiO$_2$ photoanodes, respectively (Supplementary Fig. 34). The lower work function value of hybrid PEI/TiO$_2$ indicates the reduction of TiO$_2$, which coincides with the results obtained from EELS analysis.

We assumed that the increase of Ti$^{3+}$ state could affect the electronic structure of the a-TiO$_2$ state. Therefore, valence state XPS measurement was carried out to investigate the distinct structure of hybrid PEI/TiO$_2$ compared to normal a-TiO$_2$. BiVO$_4$/TiO$_2$ exhibited a valence band composed of O 2p orbitals, and a defect state was observed below 1.5 eV from the Fermi level, with a width of 0.75 eV (Supplementary Fig. 35a−b). These values coincide with the valence band and leaky state reported in previous studies[7,32]. However, BiVO$_4$/PEI/TiO$_2$ showed a widened defect which was twice as broad as that in normal a-TiO$_2$ (Supplementary Fig. 35c−d). We conclude that the wide defect state is due to a reduction of Ti$^{4+}$ by incorporated nitrogen. The long absorption tail observed in UV-Vis spectroscopy also supports the presence of the reduced Ti$^{3+}$ state, as suggested in previous reports

(Supplementary Fig. 36)[30,33,34]. Based on these results, we propose that the partial reduction of Ti$^{4+}$ state in hybrid PEI/TiO$_2$ widens the defect band, thereby enhancing conductivity through increased density of states (DOS) in TiO$_2$.

## Charge carrier dynamics of the hybrid PEI/TiO$_2$

In previous studies, PEI polyelectrolyte has been reported to lower the work function of semiconductors[15], and it could be reasoned that a reduced work function of BiVO$_4$ contributes to the formation of a hole-selective contact by enabling favorable band bending. However, the KPFM measurement revealed that there is a negligible difference in the work function of the BiVO$_4$ photoelectrodes regardless of the thickness of the PEI layer (Supplementary Fig. 37). This result suggests that an alternative mechanism may be operative within our system.

We therefore investigated the carrier dynamics of the BiVO$_4$ photoanode and hybrid PEI/TiO$_2$ through dual-working electrode (DWE) analysis to evaluate hole transfer efficiency. We carried out operando open-circuit potential (OCP) measurement with the DWE to determine the energetics of the majority carriers of BiVO$_4$ and TiO$_2$ under dark and light conditions. In the measurement, BiVO$_4$/TiO$_2$ showed identical Fermi levels for BiVO$_4$ and TiO$_2$ under dark conditions, indicating Fermi level equilibration between the two semiconductor materials (Fig. 6a)[19,35,36]. We also observed a slight shift in the Fermi level of BiVO$_4$ under illumination due to the accumulation of photogenerated electrons, and the Fermi level of BiVO$_4$ decreased and returned to equilibrium with TiO$_2$ after turning off the light. However, the photoelectrodes modified with the PEI interfacial layer revealed a significant difference in Fermi level equilibrium compared to BiVO$_4$/TiO$_2$. The Fermi level of BiVO$_4$ did not equilibrate with that of TiO$_2$, indicating that the insulating interfacial PEI layer hinders electron exchange (majority carriers) between the two semiconductors, and also exhibited a more significant increase than BiVO$_4$/TiO$_2$ under light conditions (Fig. 6b, c). Furthermore, we observed a prolonged decay of the Fermi level energy of BiVO$_4$/PEI/TiO$_2$ after turning off the light in DWE and KPFM measurements (Fig. 6b and Supplementary Fig. 38),

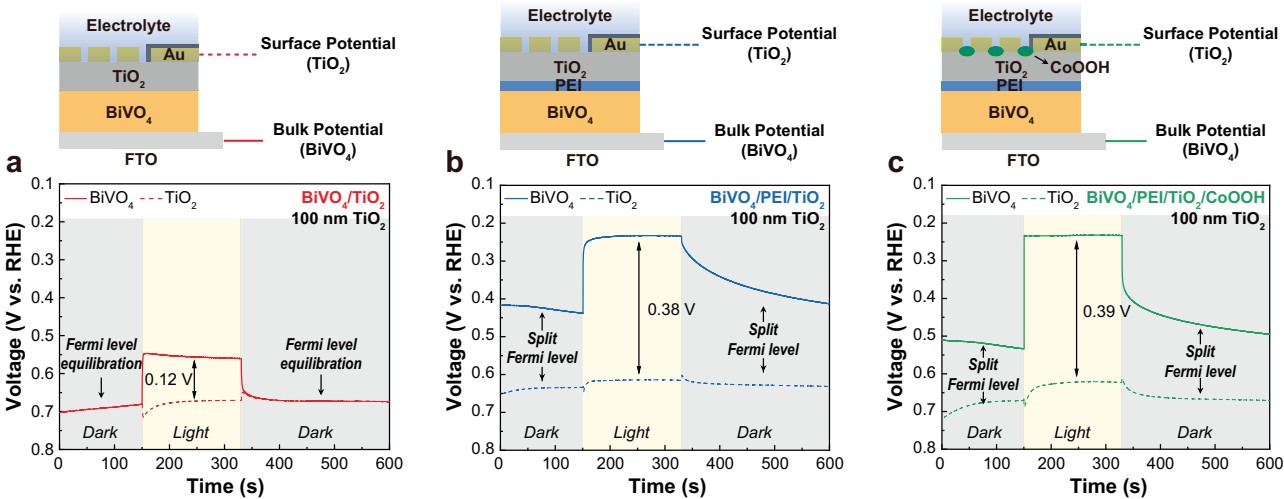

**Fig. 6 | Operando DWE analysis for investigation of charge carrier dynamics. a–c** OCP potential of BiVO₄/TiO₂ (**a**), BiVO₄/PEI/TiO₂ (**b**), and BiVO₄/PEI/TiO₂/CoOOH (**c**) detected by DWE under dark and light conditions. The schematic illustrations above each graph depict the photoanode configurations used for DWE analysis.

again indicating the electron-blocking nature of the interfacial PEI layer. The tunneling efficiency of electrons (or holes) is a function of the barrier height[37,38], and in our system the electrons in the conduction band of BiVO₄ experience a larger barrier height than the holes in the valence band due to the energetic position of the highest occupied molecular orbital (HOMO) and the lowest unoccupied molecular orbital (LUMO) levels of the thin interfacial PEI layer[39,40]. We therefore achieve selective hole transfer via tunneling across the thin interfacial PEI layer.

In the DWE and KPFM analyses, although we confirmed that selective hole transfer is facilitated by the large electron barrier height of the interfacial PEI layer, we were concerned the oxidation of PEI by photogenerated holes contributed by the small hole barrier height. For instance, a sulfite oxidation measurement revealed that BiVO₄/PEI/TiO₂ shows a transient in the photocurrent even for the expected fast reaction kinetics of the sacrificial electron donor, while it disappeared in the presence of a CoOOH co-catalyst (Supplementary Fig. 39). This result implies either low catalytic activity or slow hole extraction of the hybrid PEI/TiO₂, leading to the oxidation of the interfacial PEI layer during water oxidation, as we mentioned earlier. Therefore, we first conducted electrochemical impedance spectroscopy (EIS) for a comparative analysis between the BiVO₄/PEI/TiO₂ and BiVO₄/PEI/TiO₂/CoOOH electrodes to evaluate the catalytic activity of the hybrid PEI/TiO₂ in water oxidation (Supplementary Fig. 40). The Nyquist plot was fitted by employing a series 3RC-equivalent circuit to determine the resistance and capacitance values in each frequency domain (Supplementary Fig. 41 and 42)[19,41]. BiVO₄/PEI/TiO₂ and BiVO₄/PEI/TiO₂/CoOOH exhibited nearly identical resistance and capacitance values in the high (HF) and medium-frequencies (MF), but we observed a significant difference of resistance in the low-frequency (LF) related to the charge transfer resistance into the electrolyte and therefore the water oxidation kinetics. While BiVO₄/PEI/TiO₂ maintained a higher resistance even at high applied potential, the resistance of BiVO₄/PEI/TiO₂/CoOOH decreased steeply after the onset potential (0.28 V vs. RHE) (Supplementary Fig. 40a–b). This result shows that BiVO₄/PEI/TiO₂ has relatively low catalytic activity, emphasizing the need for an effective cocatalyst on the hybrid PEI/TiO₂ layer to achieve stable solar water oxidation. Our findings were confirmed in the stability test of BiVO₄/PEI/TiO₂ without co-catalyst. In the CA measurement, we observed that the current density of BiVO₄/PEI/TiO₂ reached nearly zero after 8 h (Supplementary Fig. 43), despite the surface morphology of the electrode remaining intact (Supplementary Fig. 44). In addition, XPS depth profiling measurements of the hybrid PEI/TiO₂ revealed the presence of

nitrogen species within the TiO₂ structure (Supplementary Fig. 45). This result points to the fact that the primary cause is the decomposition of the interfacial PEI layer. The decomposition is likely due to slow hole extraction influenced by the absence of co-catalyst, as BiVO₄/PEI/TiO₂ demonstrated long-term stability with CoOOH. The slow hole extraction leads to changes or degradation of PEI at the interface between BiVO₄ and TiO₂, hindering selective hole transfer in the BiVO₄/PEI/TiO₂. Consequently, future studies should focus on incorporating more effective co-catalysts or replacing PEI with more stable alternatives to further enhance the hole extraction within the interfacial PEI layer.

Based on our characterization and analysis, we elucidate the underlying mechanism of the selective hole transfer mediated by the interfacial PEI and the hybrid PEI/TiO₂. In the absence of the PEI interfacial layer, an unfavorable band alignment between the metal oxide and a-TiO₂ leads to sluggish hole transfer, rendering PEC water oxidation unfeasible (Fig. 7a). The introduction of the interfacial PEI layer and the hybrid PEI/TiO₂ not only mitigates this energetic misalignment but also promotes hole-selective transfer, facilitated by the electron-blocking nature of the PEI and the introduction of new defect states within the hybrid PEI/TiO₂ layer (Fig. 7b and Supplementary Fig 46).

In summary, we investigated hybrid PEI/TiO₂ as a protection layer for metal oxide photoanodes that significantly enhances the stability of these materials for solar water oxidation. The hybrid TiO₂ layer was formed on the interfacial PEI layer during the ALD process, and we observed distinct energetics associated with a new defect state resulting from the partial reduction of Ti⁴⁺ in TiO₂. Furthermore, the unique properties of the PEI layer (e.g., preventing unfavorable band bending) facilitated selective-hole transfer between the metal oxide and TiO₂ layer. Addressing the challenges related to unfavorable hole-selective contact, the metal oxide photoanode modified with hybrid TiO₂ demonstrated feasibility in solar water oxidation with long-term stability.

## Methods
### Materials
Tin(II) chloride dihydrate (SnCl₂·2H₂O), isopropyl alcohol, bismuth(III) nitrate pentahydrate (Bi(NO₃)₃·5H₂O), vanadyl acetylacetonate (VO(acac)₂), methanol, iron(III) chloride hexahydrate (FeCl₃·6H₂O), sodium nitrate (NaNO₃), polyethyleneimine (branched), hydrochloric acid (HCl), Tetrakis(dimethylamino)titanium (TDMAT), iron(II) sulfate heptahydrate (FeSO₄·7H₂O), nickel(II) nitrate hexahydrate (Ni(NO₃)₂·6H₂O), cobalt(II) nitrate hexahydrate (Co(NO₃)₂·6H₂O), sodium hydroxide (NaOH), and potassium phosphate monobasic were

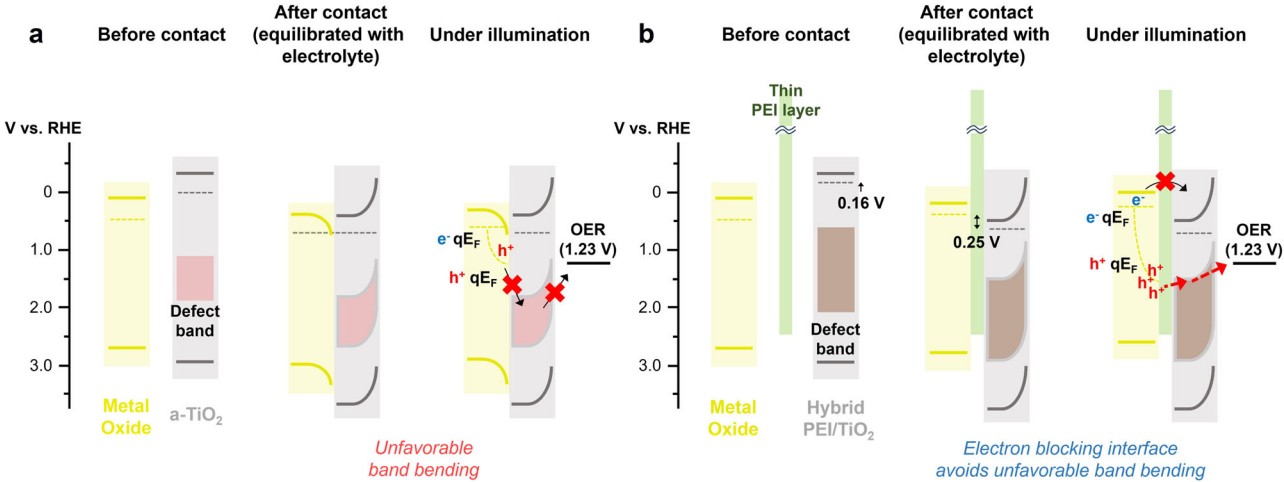

**Fig. 7 | Schematic illustration of proposed band energetics. a, b** The band energetics for metal oxide/TiO$_2$ (**a**) and metal oxide/PEI/TiO$_2$ (**b**). The band positions of the metal oxide are based on a BiVO$_4$ photoanode. The Fermi level of semiconductors and defect band position were obtained by DWE, KPFM, and valence state XPS analysis.

purchased from Sigma-Aldrich. All chemicals were used as received without further purification.

## Fabrication of BiVO$_4$ photoanode

BiVO$_4$ was fabricated on a fluorine-doped tin oxide (FTO) substrate with SnO$_2$ hole blocking layer[42]. First, 0.1 M of SnCl$_2 \cdot$2H$_2$O (0.226 g, 1.00 mmol) was dissolved in 10 mL of isopropyl alcohol with stirring for 1 h and kept for 1 day under ambient conditions. A SnO$_2$ layer was then spin-coated onto a cleaned FTO substrate at 3000 rpm for 20 s, followed by annealing at 500 °C for 1 h under ambient air conditions. BiVO$_4$ was synthesized on the SnO$_2$-coated FTO substrate using a metal-organic deposition method. A 0.5 M of Bi(NO$_3$)$_3 \cdot$5H$_2$O (1.213 g, 2.50 mmol) in 4.8 mL of acetic acid and a 0.27 M of VO(acac)$_2$ (0.357 g, 1.35 mmol) in 5 mL of methanol (aged for 3 days) were prepared, respectively. After dissolving each precursor in solvents, the V solution was added to the Bi solution with 1.1 Bi/V molar ratio (e.g. 2.961 mL of Bi solution was mixed with 5 mL of V solution). The mixed precursor solution was spin-coated on the substrate at 2000 rpm for 20 s, and the annealing process was carried out at 480 °C for 20 min under air condition. The spin-coating process was repeated to obtain the desired thickness of BiVO$_4$.

## Fabrication of Fe$_2$O$_3$ photoanode

Sn-doped Fe$_2$O$_3$ was synthesized by hydrothermal method on cleaned FTO substrate[43]. Briefly, 20 mL of precursor solution (D.I) containing 0.15 M FeCl$_3 \cdot$6H$_2$O (0.8109 g, 3.00 mmol) and 1.0 M NaNO$_3$ (1.70 g, 20.0 mmol) was transferred to Teflon-lined stainless-steel autoclave (50 mL of volume). FTO was placed at the bottom of the autoclave with conductive side facing up, and heat-treatment was conducted at 95 °C for 4 h to form β-FeOOH nanowires on the FTO substrate. After heating process, β-FeOOH was sintered in ambient air condition at 550 °C for 2 h (ramping rate of 4 K min$^{-1}$) and then annealed at 800 °C for 20 min.

## Deposition of polyethyleneimine interfacial layer

Polyethyleneimine (PEI) layer was deposited by spin-coating method on metal oxide photoanodes. PEI (Mw ~25,000) was dissolved in D.I as various concentration (wt.%) to control a thickness of PEI layer (e.g. to prepare a 4% solution, 0.2 g of PEI was dissolved in 4 ml of D.I), and pH of PEI solution was adjusted as 4.2 by 1 mL of 3 M HCl solution. The PEI solution was spin-coated on photoanodes at 4000 rpm for 20 s, followed by drying in oven at 70 °C for 30 min. The concentration and the spin-coating rate could be modulated to adjust the thickness of PEI layer.

## Deposition of amorphous TiO$_2$ layer on photoanode

Amorphous TiO$_2$ was deposited on the photoanodes by atomic layer deposition (ALD) technique (R200, Picosun). Tetrakis(dimethylamino) titanium (TDMAT, 99.99%, Aldrich) and H$_2$O were used as precursors for Ti and O, respectively. Each photoanode was placed in ALD chamber at 120 °C, TDMAT heated at 85 °C was put with a 1.6s pulse, followed by N$_2$ purge with a 6.0 s. H$_2$O was kept at room temperature with a 0.1 s pulse, followed by a 6.0 s N$_2$ purge. The thickness of TiO$_2$ layer was confirmed with Si substrate using alpha-SE ellipsometer (J.A. Woollam Co.), and 0.55 Å of TiO$_2$ layer was formed per TDMAT-H$_2$O cycle, approximately.

## Deposition of co-catalysts on the photoanodes

For deposition of FeOOH, NiOOH, and CoOOH co-catalysts, 10 mM of iron(II) sulfate heptahydrate, nickel(II) nitrate hexahydrate, or cobalt(II) nitrate hexahydrate were dissolved in D.I water. The pH was adjusted to 4.5 for the iron solution and 7.3 ~ 7.4 for the nickel and cobalt solution using 0.1 M NaOH. The photoelectrodes are soaked in the precursor solution for 3.5 h and then washed by D.I and dried by N$_2$ gun, gently.

## Deposition of metal co-catalyst on the photoanodes

For a BiVO$_4$/TiO$_2$ photoanode, 5 nm of metal layers was deposited onto the photoanode surface by Leica EM ACE600 magnetron sputter. Ni (100 mA working current and 2.0 E$^{-2}$ mbar), Pt (35 mA working current and 5.0 E$^{-2}$ mbar), and Au (30 mA working current and 5.0 E$^{-2}$ mbar) were deposited under certain conditions.

## Photoelectrochemical Characterizations

Photoelectrochemical (PEC) characterizations were carried out by SP-200 Bio-Logic potentiostat in a three-electrode configuration under AM 1.5G illumination. the photoanode, Pt wire, and Ag/AgCl electrode were used as working, counter, and reference electrode, respectively. For the measurement of PEC performance, 0.5 M potassium phosphate (KPi) was used as the electrolyte under back-side (BiVO$_4$) and front-side (Fe$_2$O$_3$) illumination, and epoxy resin (Loctite Epoxide-resin EA 9461 and EA 9466) was used to obtain certain surface area of the photoanodes (0.2 ~ 0.25 cm$^{-2}$). IPCE measurement was conducted using a home-built double mono-chromator with a halogen light source. The light intensity of the monochromator was calibrated using a Si diode. Electrochemical impedance spectroscopy (EIS) was measured by SP-300 with 10 mV AC voltage amplitude and frequency range from 0.2 Hz to 1 MHz

under the illumination of white light LED (SP-12-W5, cool white Luxeon Rebel). Numerical fitting of EIS data was conducted by Zview software.

## Characterizations

A surface and cross-sectional morphology were analyzed with a Hitachi SU-7000 field-emission scanning electron microscope (SEM) and Zeiss Gemini 450 SEM. The Bi and V contents in electrolytes after stability test were evaluated with a Varian inductively coupled plasma-optical emission spectrometer (ICP-OES). High-resolution X-ray photoelectron spectroscopy (XPS) spectra of the photoanodes was obtained by Thermo Fisher K-Alpha XPS instrument, and valence state XPS analysis to confirm leaky state was carried out with Physical Electronics (PHI) Quantum 2000 X-ray photoelectron spectrometer featuring monochromatic Al-Kα radiation, generated from an electron beam operated at 15 kV and 35 W. The energy scale linearity of the instrument was established through calibration with a reference sample of Au. Time-of-flight secondary ion mass spectrometry (TOF-SIMS) and X-ray crystallography (XRD) were conducted to investigate the depth-profiling and crystallinity of photoanodes through IONTOF TOF-SIMS-5 and Rigaku Smartlab diffractometer, respectively. The elemental composition in the cross-sectional direction was obtained by high-resolution transmission electron microscopy (HRTEM) with a JEOL JEM 2010 transmission electron microscope. Electron energy-loss spectroscopy (EELS) was used to compare oxidation state, and the measurement was carried out using a Cs-corrected TEM with JEOL JEM-ARM300 transmission electron microscope to ensure high-resolution. HAADF-STEM and EELS resolution are 0.058 nm and 0.3 eV with 300 kV acceleration voltage, respectively. The samples for TEM measurement were prepared by focused ion beam (FIB) milling. The absorbance measurement was carried out by UV-Vis spectroscopy with CRAIC 20/20/PV UV-Vis microspectrometer.

## Kelvin probe force microscopy measurement

Kelvin probe force microscopy (KPFM) was carried out to confirm work function (WF) of each photoanode. An Asylum Research AFM (MFP-3D) was used to measure the work function of the samples. The probe used for the measurement was a AC240TM-R3. For calibration of the work function of the tip a highly ordered pyrolytic graphite (HOPG) has been used which has a reported work function of ~ 4.6 eV[44]. To achieve a fresh HOPG surface a piece of scotch tape was used to pull off a few top layers of the graphite and exposing a fresh clean surface for the calibration. The HOPG used was purchased from MikroMasch (Grade: ZYA). The fresh HOPG surface changes its work function in a time window of several tens of minutes when exposed to air. Therefore, the HOPG was measured against an Aluminium metal mirror with native $Al_2O_3$ layer on the surface. The work function of the $Al/Al_2O_3$ (~3.90 eV) was stable for several hours and even days[45]. Open-source Gwyddion software package as well as the Asylum Research build in software were used to further analyze the AFM pictures and determine the average work function of the surface.

## Dual-working electrode measurement

Initially, a 10 nm-thick permeable Au layer was deposited using sputtering onto each photoanode surface, which had been covered with epoxy resin (Epoxide-resin EA 9461, Loctite). To establish an additional contact for surface potential measurement of the $TiO_2$ layer, a front contact was fabricated on the Au layer of the epoxy surface. The front contact was connected with a Cu foil and the Au layer using Ag paste and sealed by additional epoxy to protect the contact from the electrolyte. Open circuit potential (OCP) measurement was carried out using a dual-working electrode (DWE) through Bio-Logic SP-300 potentiostat with V2-controlled PEIS measurements. A first and second

working electrode (WE1 and WE2) were connected to the $BiVO_4$ back contact and the $TiO_2$ front contact, allowing to monitor changes in Fermi level and quasi electron Fermi level under dark and light illumination.

## Data availability

The authors declare that the data supporting the findings of this study are available within the paper and its supplementary information files, including the source data file. Source data are provided with this paper.

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

## Acknowledgements
This work was supported by the University Research Priority Program LightChEC of the University of Zurich, the Swiss National Science Foundation (Project #184737 and #208451), the Basic Science Research Program (2021R1A2C2013684), and the Regional Leading Research Center (RLRC) (RS-2023-00217778) funded by the National Research Foundation (NRF) of Korea.

## Author contributions
S.B., J.R. and S.D.T. conceived the concept. S.B. performed the materials design, characterization, and photoelectrochemical experiments and analyses. T.M. carried out KPFM analysis. T.M. and E.S. performed EIS analysis. M.K and Y.C. conducted TOF-SIMS measurement. P.A. and Z.W. performed XPS and XRD analysis. S.B., J.R. and S.D.T. wrote the manuscript. All authors discussed the results and participated in writing the manuscript.

## Competing interests
The authors declare no competing interests.
