## [Peer Review File · Nature Communications]

A Hole-Selective Hybrid TiO₂ Layer for Stable and Low-Cost Photoanodes in Solar Water Oxidation

Corresponding Author: Professor S. Tilley

Version 0:

Reviewer comments:

Reviewer #1

(Remarks to the Author)

no significant technological novelties are reported in this manuscript. Neither from the point of view of scientific knowledges Therefore, I cannot recommend this manuscript for its consideration in Nature Communications.

Nowasays, "These PEI/TiO₂ modified photoanodes exhibit high photostability for solar water oxidation (~2 mA cm⁻² for > 120 h)" is a not relevant scientific or technological conclusion. Higher corrent density are required to define an enough competitive phottoelectrochemical system.

Neither new insighths about the hole transfer mechanisms are discussed. Degradation analysis during 120hours is not a worthy aging anlysis.

Reviewer #2

(Remarks to the Author)

The manuscript entitled "Hole-Selective Hybrid TiO₂ Layer towards Long Term Stability of Low-Cost Photoanodes in Solar Water Oxidation" presents a novel method to improve the photostability of photoanodes. The authors discuss the development of hybrid polyethyleneimine (PEI)/TiO₂ layers that enhance the performance and stability of metal oxide photoanodes used in solar water oxidation. The key innovation lies in the integration of a thin PEI interlayer between the photoanode and the TiO₂ protective overlayer. This configuration not only protects the underlying photoanode from photodegradation but also facilitates effective hole transfer, improving the optoelectronic properties of the devices. The content of the manuscript is generally systematic and logical, and the elucidation of the mechanism makes sense. This manuscript can be further considered by Nat. Commun. subject to a major revision, along the lines suggested hereafter:

1. To ensure that readers fully understand the significance and role of PEI in this study, the author should provide a detailed background of PEI in the introduction section. This includes the chemical properties of PEI, its common applications in photoelectric materials, and the specific advantages of using PEI as an interface layer in this research.
2. The authors should explain their choice of CoOOH co-catalysts over alternatives such as NiOOH or FeOOH co-catalysts.
3. The authors need to present evidence confirming the successful deposition of co-catalysts on the photoanodes.
4. The authors should provide details of the changes in the photoanodes before and after stability testing. These changes include alterations in morphology, structural composition, and valence states.
5. The authors should provide the Applied Bias Photon-to-current Efficiency (ABPE) and Incident Photon-to-Current Efficiency (IPCE) of the photoanodes, which are crucial metrics in evaluating the performance of photoelectrode materials.
6. Authors are requested to add and compare the state-of-art reported photoanodes with good stability, including BiVO₄ photoanodes, Fe₂O₃ photoanodes, Si-photoanodes, and Ta₃N₅ photoanodes.

Reviewer #3

(Remarks to the Author)

In this study, Bea et al. have investigated the origin of the stability improvement of most commonly used oxide photoanodes (BiVO₄ and Fe₂O₃) via PEI/TiO₂ coating. Coating strategies have previously been suggested to protect the photocorrosion of oxide photoanodes, TiO₂ has been the most common choice of coating overlayer. They have presented a set of experiments and demonstrated that PEI/TiO₂ facilitates hole transfer, despite the surface has low water oxidation activity. Before recommending this paper for publication, the following must be addressed.

100 nm thick TiO₂ overlayers on bare oxide photoanodes act as a hole blocking layer due to the lower band edge, in reversed type II heterojunction configuration. Later, the authors show this in the band diagrams. Therefore, it is not surprising that to get a low photocurrent. It becomes leaky when the overlayer is not uniform, contains pinholes. So, there is many other reasons for "leaky" performance.

TOF-SIMS analysis shows a significant amount of carbon in BiVO₄/TiO₂ system. The authors acknowledge that the source of carbon is the ALD precursor Ti. TiO₂ has been considered amorphous, but what is the evidence that the overlayer is actually TiO₂, rather than Ti oxide with C mixed matrix?

In LSV scans under chopped illumination, there is a cathodic peaks at a low potential range once PEI/TiO₂ is present. What would be the origin of this behaviour? It seems like the appearance and disappearance of these cathodic features is correlated with the PEI thickness and therefore the overall photoactivity.

In the case of BiVO₄, the back illumination has been performed. What would be the case once front illumination is performed?

The improved activity has been attributed to the presence of Ti³⁺ species in the PEI matrix. Energy differences in Ti L edge EEL spectra and the noisy VB spectra have been shown as the evidence. The authors must confirm this by comparing Ti 2p XPS spectra. It will be less noisy and Ti³⁺ peak, if exists, would be apparent. Also, the peak shift in Ti L edge EEL spectra is not apparent. The authors must compare at least two spectra from TiO₂ and PEI/TiO₂ matrix, to justify the peak shift.

Judging by the SEM images presented, there is a significant difference in the morphology of the photoanodes surfaces once coated with PEI/TiO₂. What would be the reason for this? More compact and uniform layers could contribute to the overall activities.

Can authors exclude the oxidation of PEI under potential/illumination?

Reviewer #4

(Remarks to the Author)

The paper describes a new type of protection layers for photoanodes that is composed of the organic polymer polyethylene imine and ALD processed amorphous TiO₂. The system shows a surprisingly good performance that is certainly be worth communicating. However, the presented discussion of the many data collected in this paper have serious flaws in the discussion/interpretation. Data that actually do not provide a new insight might be omitted. Overall, the manuscript is considerably too long for a communication paper.

I recommend reconsideration after a complete revision.

1)

Lines 97/99 here is a comparison of current density (A/cm²) to current (A). This is not traceable without knowledge of the electrode area. Overall, this should be avoided for better readability. Same problem in Line 196.

2)

Line 106, wok function of the metal. It seems a bit questionable that under operationg condition the surface is formed by a metal. It could also be that the metal is oxide covered under the operating conditions. How did the author establish that the catalyst is in the metallic state?

3)

Line 112 "Further discussion on the origin of the photocurrent in BiVO₄/PEI/TiO₂ will be discussed in a subsequent section." [Avoid "overdiscussing" ;-)]

4)

The XPS survey spectra are not very informative and might be placed in SI. The Figure S9 should be supplemented by a peak fit. The authors should discuss the origin of an F 1s signal in Figure 2a.

In light of the discussion about involvement of PEI in oxidation/degradation, it would be interesting to analyze high resolution spectra of the as prepared electrodes and after prolonged usage.

5)

Line 183, "These 183 results suggest that variations in the measured photocurrent mainly stem from the amount of 184 embedded C and N within the hybrid PEI/TiO₂ layer." This sentence should be deleted. A correlation does not mean causality. PEI is not a crystalline compound, in which elemental composition determine crystal structure and electronic properties. It is a molecular compound whose properties depend mainly on the connectivity of atoms. In addition, SIMS (and XPS) cannot determine the content of H, which is contained in PEI at a high mole fraction. The authors may take another organic material with similar atomic N:C ratio as PEI and try the effect. Most likely such a compound would not at all functionable.

6)
Have the authors evaluated the possibility that the PEI layer modulates the pH at the interface that could have a very pronounced effect on the photocurrent? Under operating condition, the local pH will deviate from the bulk pH. PEI may support or counteract this effect.

7)
P. 15, EIS; it remains completely unclear what was actually done and how the authors can arrive at their conclusions. Which potential was externally applied while making the EIS measurement? If no potential was applied, what was the open circuit potential that is established spontaneously? Which electrocatalytic reactions run at the DC potential? If no reaction is running, how can the impedance data reveal anything about catalytic activity?

Version 1:

Reviewer comments:

Reviewer #2

(Remarks to the Author)

I am satisfied with the experiments supplemented and changes made in the manuscript, and thus this manuscript could be accepted for publication.

Reviewer #3

(Remarks to the Author)

The authors have addressed thoroughly all the questions I raised. I have also gone through their responses to the comments and questions of other reviewers. I believe that they have revised the manuscript reasonably well and in this form, this study could be considered for publication.

Reviewer #4

(Remarks to the Author)

The authors have amended their manuscript by many important data and considerably improved the quality of the discussion formally and scientifically.

While this applies to most data, the reviewer is still not satisfied with the interpretation of the XPS data.

Supplementary Figure 16 and Supplementary Figure 33b: It is clear that there is a Ti 2p signal confirming the statement on p7 of the manuscript. For this statement, Supplementary Figure 16a would fully suffice. However, Supplementary Figure 16b contains a fit of the Ti 2p_{1/2} component. This fit interprets the broad peak as indication of two Ti binding states. However, the Ti 2p_{3/2} and Ti 2p_{1/2} components must have an area ratio of 2:1. As the peak height for the Ti 2p_{1/2} component is smaller, the width necessarily must be broader without automatically implying different binding states. Different binding states should also be evident from the Ti 2p_{3/2} component. The reviewer admits that the full interpretation of Ti 2p spectra can be challenging due to suboxides and other satellites. The handling of the baseline is also not trivial. However, when attempting a fitting, it must be based on a physical basis and this should be spelled out (probably in the SI). The presented fitting implies a different spin-orbit splitting depending on the chemical state of the oxide. All this comes without literature reference or other justification, which the reviewer finds unacceptable.

While the data itself in Supplementary Figure 16a support the statement on p7, this is not the case for Supplementary Figure 33b with respect to the statement on p 13. Here all the deficits listed above directly impact on the validity of the conclusion. For this reason, the paper cannot be recommended in the present state.

As a further point, the reviewer recommends caution with destructive XPS depth profiling (Supplementary Figure 45). The chemical states of the compounds is usually *not* preserved and such conclusion should not be made or implied. A clarifying statement in this regards would help to avoid ambiguities with the readers.

Version 2:

Reviewer comments:

Reviewer #4

(Remarks to the Author)

The authors have addressed the issues with the XPS data evaluation and I consider the material publishable now. As Nat. Commun. considers itself as a prime journal, the authors should be asked to flatten out the remaining inaccuracies introduced with the sentences added to the revised manuscript. I consider that an important but technical formality and do not want to see the manuscript again.

SI, Figure caption 33: „To confirm the peak proportion, peak fitting was carried out under the following conditions: Smoothing of the raw data, Shirley background subtraction with an average width of 1, an area constraint ratio of 1:2 between 2p_{1/2} and 2p_{3/2}, and position constraints of 5.7 for the Ti⁴⁺ and Ti³⁺ peaks in TiO₂.“

- 1) The quantities „Average width“ and „position constrains“ lack a unit.
- 2) The term „position constrain“ is vague and not understandable to a wider audience. From the context, the reviewer speculates that the authors refer here to the very established technical term „spin-orbit splitting“. The following reading source may be recommended on this: P. v. der Heide; X-ray Photoelectron Spectroscopy. Wiley & Sons, Hoboken, 2012, pp. 108-111.

Response to Reviewers' Comments

(NCOMMS-24-16729-T)

We thank the reviewers for their comments, which have helped to improve the quality and clarity of this manuscript. Below is our point-by-point response to the Reviewers' comments. Our response to the comments and any changes made to the original manuscript were highlighted in blue and green, respectively.

Reviewer #1

Comments to the Author:

No significant technological novelties are reported in this manuscript. Neither from the point of view of scientific knowledges. Therefore, I cannot recommend this manuscript for its consideration in Nature Communications. Nowadays, "These PEI/TiO₂ modified photoanodes exhibit high photostability for solar water oxidation (~2 mA cm⁻² for > 120 h)" is a not relevant scientific or technological conclusion. Higher current density are required to define an enough competitive photoelectrochemical system. Neither new insights about the hole transfer mechanisms are discussed. Degradation analysis during 120hours is not a worthy aging analysis.

Author's response:

We thank the reviewer for the time invested in evaluating our manuscript. We would like to address the concerns that have been raised regarding the novelty and significance of our research.

(1) Novelty

The main impact of this work is the discovery of an easily deposited protective layer for water splitting photoanodes that is selective for holes, which to the best of our knowledge is unprecedented. The importance of selective contacts in maximizing the efficiency of solar energy conversion in water splitting is only recently being recognized, though it has long been appreciated in photovoltaics. While "leaky" TiO₂ has previously been demonstrated to conduct holes injected from very high quality (and expensive) semiconductors (though no evidence was given for selectivity of the contact: *Science* **2014**, *344*, 1005-1009 / *Energy Environ. Sci.*, **2014**, *7*, 3334–3337), a-TiO₂ by itself does not work for low-cost, emerging semiconductors, which will likely be employed in cost effective photoelectrochemical or photocatalytic particle systems. The new

protective layer introduced here therefore opens up the use of a much larger variety of materials for photoelectrochemical and photocatalytic water splitting.

(2) Photocurrent

We agree that it is often unclear how well studies with significantly lower photocurrents than typical (e.g., <5%) will translate to "high performance" systems. However, our photocurrents are only a factor of 2-3 lower than what could be considered "high" photocurrents. Moreover, there is a light intensity dependence for our samples, meaning that if one extrapolates the photocurrent under 10% sun to 1 sun, much higher photocurrents would be expected ($>4 \text{ mA cm}^{-2}$). We have observed such light intensity dependence in other systems with more typical overlayers (*ACS Catal.* **2024**, *14*, 9877–9886), so this need not be related to the PEI/TiO₂ layer itself. We argue that the photocurrents are sufficient to evaluate the main impact of the paper, which is described above.

(3) Photostability

Regarding the photostability, we agree that a longer stability test would enrich and strengthen our manuscript. In our original data (**Supplementary Fig. 6 in the original manuscript**), we carried out a long-term stability test for 250 h, but observed a decrease in photocurrent even after 50 h. We hypothesized that this was due to the detachment of co-catalyst (CoOOH) caused by oxygen bubbles. To prove the hypothesis, we redeposited the co-catalyst on the already used photoanode and conducted a further long-term stability test again, extending previous stability test to 400 h. This extended data range offers a more comprehensive demonstration of photostability, and we will incorporate this observation to strengthen our discussion. We believe that it provides a meaningful indication of the durability of the hybrid PEI/TiO₂ under PEC water oxidation condition, enhancing the manuscript's value.

Revision made (highlighted):

[Supplementary Fig. 6 in the original manuscript]

Supplementary Fig. 1. Long-term stability test of BiVO₄/PEI/TiO₂/CoOOH. CA measurement of BiVO₄/PEI/TiO₂/CoOOH in 0.5 M KPi electrolyte (pH 8) at 1.23 V vs. RHE.

[Supplementary Fig. 13 in the revised manuscript]

Supplementary Fig. 13. Long-term stability test of BiVO₄/PEI/TiO₂/CoOOH. CA measurement of BiVO₄/PEI/TiO₂/CoOOH in 0.5 M KPi electrolyte (pH 8) at 1.23 V vs. RHE. After 250 h stability test, the cobalt co-catalyst was redeposited, followed by an additional 150 h stability test.

Reviewer #2

Comments to the Author:

The manuscript entitled “Hole-Selective Hybrid TiO₂ Layer towards Long Term Stability of Low-Cost Photoanodes in Solar Water Oxidation” presents a novel method to improve the photostability of photoanodes. The authors discuss the development of hybrid polyethyleneimine (PEI)/TiO₂ layers that enhance the performance and stability of metal oxide photoanodes used in solar water oxidation. The key innovation lies in the integration of a thin PEI interlayer between the photoanode and the TiO₂ protective overlayer. This configuration not only protects the underlying photoanode from photodegradation but also facilitates effective hole transfer, improving the optoelectronic properties of the devices. The content of the manuscript is generally systematic and logical, and the elucidation of the mechanism makes sense. This manuscript can be further considered by Nat. Commun. subject to a major revision, along the lines suggested hereafter:

Comment #2-1:

To ensure that readers fully understand the significance and role of PEI in this study, the author should provide a detailed background of PEI in the introduction section. This includes the chemical properties of PEI, its common applications in photoelectric materials, and the specific advantages of using PEI as an interface layer in this research.

Author's response:

We appreciate the reviewer's valuable comments. Following the reviewer's suggestion, we described a detailed information about PEI in the introduction section.

Revision made (highlighted):

[Line 69, page 4 in the revised manuscript]

In this context, polyethyleneimine (PEI), which consists of repeating units of amine groups and two carbon aliphatic spacers, was selected to alleviate unfavorable interface energetics and facilitate hole-selective transfer. The non-conjugated PEI is known not only as a modifier of work function^{15,16} but also as a hole transfer channel^{17,18} since the amine moieties are easily oxidized, enabling hole transfer in PEC devices. We considered these unique properties of PEI to be promising for effectively addressing the aforementioned issues at the interface between metal

oxides and α -TiO₂.

Comment #2-2:

The authors should explain their choice of CoOOH co-catalysts over alternatives such as NiOOH or FeOOH co-catalysts.

Author's response:

We explored surface co-catalysts that can be deposited by a very simple immersion process, which is preferable for translatability to related systems such as photocatalytic particles or particle sheets. The immersion process enables the formation of the self-assembled co-catalysts on the hybrid PEI/TiO₂ layer, thereby preserving its integrity and functionality. For surface co-catalysts deposited by the immersion process, CoOOH is superior to FeOOH and NiOOH (**Supplementary Fig. 9 in the revised manuscript**).

Revision made (highlighted):

[**Supplementary Fig. 9 in the revised manuscript**]

Supplementary Fig. 9. PEC water oxidation performance of BiVO₄/PEI/TiO₂ with various co-catalysts. LSV curves of BiVO₄/PEI/TiO₂ with FeOOH (a) and NiOOH (b) co-catalysts. BiVO₄/PEI/TiO₂/CoOOH (c) exhibited superior PEC performance in terms of onset potential and photocurrent compared to the other co-catalysts.

[**Line 123, page 6 in the revised manuscript**]

We also prepared co-catalyst modified photoanodes (BiVO₄/CoOOH and BiVO₄/PEI/TiO₂/CoOOH) to facilitate efficient water oxidation (**Supplementary Fig. 7 and 8**).

The CoOOH was selected due to its superior catalytic activity compared to FeOOH and NiOOH deposited via the immersion process (**Supplementary Fig. 9**).

[**Line 388, page 20** in the revised manuscript]

Deposition of co-catalysts on the photoanodes

For deposition of FeOOH, NiOOH, and CoOOH co-catalysts, 10 mM of iron(II) sulfate heptahydrate, nickel(II) nitrate hexahydrate, or cobalt(II) nitrate hexahydrate were dissolved in D.I. water. The pH was adjusted to 4.5 for the iron solution and 7.3 ~ 7.4 for the nickel and cobalt solution using 0.1 M NaOH. The photoelectrodes were soaked in the precursor solution for 3.5 h, washed with D.I. water and then dried under a gentle stream of N₂ gas.

Comment #2-3:

The authors need to present evidence confirming the successful deposition of co-catalysts on the photoanodes.

Author's response:

Thank you, we have now included XPS analysis of BiVO₄/PEI/TiO₂/CoOOH to confirm the presence of cobalt.

Revision made (highlighted):

[**Supplementary Fig. 8** in the revised manuscript]

Supplementary Fig. 8. XPS analysis of BiVO₄/PEI/TiO₂/CoOOH. High resolution XPS spectra of BiVO₄/PEI/TiO₂/CoOOH. XPS analysis showed a Co 2p peak around 782.2 eV.

Comment #2-4:

The authors should provide details of the changes in the photoanodes before and after stability testing. These changes include alterations in morphology, structural composition, and valence states.

Author's response:

Following the reviewer's suggestion, we analyzed the changes in the photoanode characteristics after the 120 h chronoamperometry (CA) measurement. First, SEM measurements revealed no significant differences in morphology compared to the photoanode before the stability test, and the hybrid PEI/TiO₂ layer remained intact on the BiVO₄ surface. High resolution XPS spectra and depth profiling analysis also showed a clear Ti 2p peak, including Ti³⁺ state, and N 1s peak within the structure of the hybrid PEI/TiO₂ (~80 nm) even after 120 h stability test. These results suggest that the hybrid PEI/TiO₂ could be used as a stable protection layer for PEC water oxidation.

Revision made (highlighted):

[Supplementary Fig. 15 in the revised manuscript]

Supplementary Fig. 15. SEM image of BiVO₄/PEI/TiO₂/CoOOH after the stability test. a-b, SEM measurements revealed no significant change in the surface morphology even after 120 h stability test.

[Supplementary Fig. 16 in the revised manuscript]

Supplementary Fig. 16. XPS analysis of BiVO₄/PEI/TiO₂/CoOOH after the stability test. a-b, High resolution XPS spectra of BiVO₄/PEI/TiO₂/CoOOH after 120 h stability test. XPS analysis showed a clear Ti 2p peak including Ti³⁺ state in BiVO₄/PEI/TiO₂/CoOOH.

[REDACTED]

[Line 141, page 7 in the revised manuscript]

The SEM measurement revealed negligible changes in morphology compared to the photoanode before the stability test (**Supplementary Fig. 15**), and high resolution XPS spectra showed a clear Ti 2p peak in BiVO₄/PEI/TiO₂/CoOOH even after the 120 h stability test (**Supplementary Fig. 16**).

Comment #2-5:

The authors should provide the Applied Bias Photon-to-current Efficiency (ABPE) and Incident Photon-to-Current Efficiency (IPCE) of the photoanodes, which are crucial metrics in evaluating the performance of photoelectrode materials.

Author's response:

Following the reviewer's suggestion, we conducted ABPE and IPCE analyses. The half-cell ABPE analysis showed maximum efficiencies of 0.15%, 0.61%, and 0.73% for the BiVO₄, BiVO₄/CoOOH, and BiVO₄/PEI/TiO₂/CoOOH, respectively. In the IPCE measurements, we evaluated the photoconversion efficiency of BiVO₄, BiVO₄/PEI/TiO₂, and BiVO₄/PEI/TiO₂/CoOOH. Interestingly, while the integrated photocurrent density from the IPCE measurements (obtained at 10% sun white light bias) of bare BiVO₄ (1.02 mA cm⁻²) was well-matched with the photocurrent density obtained in the LSV measurement, BiVO₄/PEI/TiO₂ (1.33 mA cm⁻²) and BiVO₄/PEI/TiO₂/CoOOH (4.08 mA cm⁻²) showed higher integrated photocurrent densities compared to those from the LSV. The origin of this light intensity dependence will be explored in future work.

Revision made (highlighted):

[Supplementary Fig. 10 in the revised manuscript]

Supplementary Fig. 10. ABPE of BiVO₄, BiVO₄/CoOOH and BiVO₄/PEI/TiO₂/CoOOH. The half-cell ABPE values was converted from photocurrent densities of each photoanode in the LSV curve to determine a thermodynamically based conversion efficiency at a given applied potential.

[Supplementary Fig. 11 in the revised manuscript]

Supplementary Fig. 11. IPCE of BiVO₄, BiVO₄/PEI/TiO₂, and BiVO₄/PEI/TiO₂/CoOOH. The IPCE measurement was carried out in 0.5 M KPi (pH 8) at 1.23 V vs. RHE under 10% white light illumination. The integrated photocurrent density of BiVO₄ (1.02 mA cm⁻²) matched well with that from the LSV curve, while BiVO₄/PEI/TiO₂ (1.33 mA cm⁻²) and BiVO₄/PEI/TiO₂/CoOOH (4.08 mA cm⁻²) exhibited higher integrated photocurrent densities compared to their LSV measurements, which indicates a limitation at higher light intensities.

[Supplementary Fig. 12 in the revised manuscript]

Supplementary Fig. 12. Light intensity dependence of the photocurrent in BiVO₄/PEI/TiO₂/CoOOH. Chronoamperometry (CA) of BiVO₄/PEI/TiO₂/CoOOH (a) was

carried out in 0.5 M KPi (pH 8) under white light (1% ~ 100% of 1 sun illumination by white light LED). The photoanode showed a decreasing slope of photocurrent improvement with increasing light intensity (**b**), indicating limitations of the catalytic activity of CoOOH at higher light intensities.

[Line 129, page 7 in the revised manuscript]

The applied bias photon-to-current efficiency (ABPE) showed maximum efficiencies of 0.15%, 0.61%, and 0.73% for the BiVO₄, BiVO₄/CoOOH, and BiVO₄/PEI/TiO₂/CoOOH, respectively (**Supplementary Fig. 10**). We also evaluated the photoconversion efficiency of BiVO₄, BiVO₄/PEI/TiO₂, and BiVO₄/PEI/TiO₂/CoOOH electrodes through the incident photon-to-current efficiency (IPCE) measurement (**Supplementary Fig. 11 and 12**). In the measurement, the photoanodes with the overlayers exhibited much higher conversion efficiencies at longer wavelengths near the band gap.

[Line 408, page 21 in the revised manuscript]

IPCE measurement was conducted using a home-built double monochromator with a halogen light source. The light intensity of the monochromator was calibrated using a Si diode.

Comment #2-6:

Authors are requested to add and compare the state-of-art reported photoanodes with good stability, including BiVO₄ photoanodes, Fe₂O₃ photoanodes, Si-photoanodes, and Ta₃N₅ photoanodes.

Author's response:

Following the reviewer's suggestion, we added a table that includes information on the stability and photocurrent density of state-of-the-art photoanodes.

Revision made (highlighted):

[Table S1 in the revised manuscript]

Table S1. The comparison of photoanode stability.

Photoanode (highlighted in bold)	Electrolyte	Voltage (vs. RHE)	Photocurrent density (mA cm⁻²)	Stability (h)	Ref
BiVO₄/PEI/TiO₂ /CoOOH	0.5 M KPi (pH 8.0)	1.23 V	2.03	400	This work
BiVO₄/NiFe(OH)_x	1 M KBi (pH 9.0)	0.6 V	1.6	250	1
BiVO₄/ZCF(P)-O	1 M KBi (pH 9.0)	0.7 V	4.0	40	2
BiVO₄/NiFeO_x /PAAM	1 M KBi (pH 9.0)	1.23 V	3.0	500	3
Mo:BiVO₄/NTO /Fe_xNi_{1-x}O	H ₃ PO ₄ /NaOH (pH 7.0)	1.23 V	5.6	16	4
BiVO₄/FeOOH /NiOOH	1 M KBi (pH 9.3)	0.6 V	2.8	60	5
Fe₂O₃/ZnO/CoTCPP/Fe OOH	1 M NaOH (pH 13.6)	1.23 V	3.07	20	6
Fe₂O₃/Fe₂TiO₅/LDH	1 M NaOH (pH 13.6)	1.23 V	3.54	20	7
Ti:Fe₂O₃/CoO_x/Ni	1 M NaOH (pH 13.6)	1.23 V	1.05	10	8
n-Si/SiO_x /CoO_x-Mo₂%	1 M KBi (pH 9.5)	1.47 V	18.0	43	9
n-Si/ZrO₂/Ni/NiO /Ir SAs	1 M NaOH (pH 13.6)	1.23 V	27.7	130	10
n-Si/a-TiO₂/Ni	1 M KOH (pH 14.0)	1.8 V	30.0	600	11
Si/GaAs NW /a-TiO₂/NiO_x	1 M KOH (pH 14.0)	1.5 V	10.0	600	12
TiN/Ta₃N₅ /CPF-TTB/NiFeO_x	1 M KOH (pH 14.0)	1.23 V	9.12	1.25	13
Al₂O₃/GaN/Ta₃N₅ /NiFeO_x	1 M KOH (pH 13.8)	1.23 V	7.37	1.5	14
In:GaN/Ta₃N₅ /Mg:GaN/NiCoFe-B_i	1 M KOH (pH 13.6)	1.0 V	9.3	2.7	15
Mg:Ta₃N₅/NiCoFe-B_i	1 M KOH	1.0 V	8.0	5	16

	(pH 13.6)				
--	-----------	--	--	--	--

- 1 Wang, S. *et al.* Decoupled crystallization and particle growth of BiVO₄ via rapid thermal process for enhanced charge separation. *Adv. Funct. Mater.* 2403019 (2024).
- 2 Pan, J. B. *et al.* Introducing bidirectional axial coordination into BiVO₄@metal phthalocyanine core-shell photoanodes for efficient water oxidation. *Angew. Chem. Int. Ed.* **62**, e202307246 (2023).
- 3 Tan, J. *et al.* Hydrogel protection strategy to stabilize water-splitting photoelectrodes. *Nat. Energy* **7**, 537-547 (2022).
- 4 Beetz, M. *et al.* Ultra-thin protective coatings for sustained photoelectrochemical water oxidation with Mo:BiVO₄. *Adv. Funct. Mater.* **31**, 2011210 (2021).
- 5 Lee, D. K. *et al.* Enhancing long-term photostability of BiVO₄ photoanodes for solar water splitting by tuning electrolyte composition. *Nat. Energy* **3**, 53-60 (2017).
- 6 Xie, H. *et al.* Engineering surface passivation and hole transport layer on hematite photoanodes enabling robust photoelectrocatalytic water oxidation. *ACS Nano* **18**, 5712-5722 (2024).
- 7 Fouemina, J. C. N. *et al.* Surface self-transforming FeTi-LDH overlayer in Fe₂O₃/Fe₂TiO₅ photoanode for improved water oxidation. *Small* **19**, e2301114 (2023).
- 8 Mao, L. *et al.* Synergy of ultrathin CoO_x overlayer and nickel single atoms on hematite nanorods for efficient photo-electrochemical water splitting. *Small* **19**, e2203838 (2023).
- 9 Peng, S. *et al.* n-Si/SiO_x/CoO_x-Mo photoanode for efficient photoelectrochemical water oxidation. *Small* **20**, e2304376 (2024).
- 10 Jun, S. E. *et al.* Atomically dispersed iridium catalysts on silicon photoanode for efficient photoelectrochemical water splitting. *Nat. Commun.* **14**, 609 (2023).
- 11 Dong, Y. *et al.* Substantial lifetime enhancement for Si-based photoanodes enabled by amorphous TiO₂ coating with improved stoichiometry. *Nat. Commun.* **14**, 1865 (2023).
- 12 Shen, X. *et al.* Defect-tolerant TiO₂-coated and discretized photoanodes for >600 h of stable photoelectrochemical water oxidation. *ACS Energy Lett.* **6**, 193-200 (2020).
- 13 Yang, J. W. *et al.* Conjugated polythiophene frameworks as a hole-selective layer on Ta₃N₅ photoanode for high-performance solar water oxidation. *Adv. Funct. Mater.* 2400806 (2024).
- 14 Higashi, T. *et al.* Design of semitransparent tantalum nitride photoanode for efficient and durable solar water splitting. *Energy Environ. Sci.* **15**, 4761-4775 (2022).
- 15 Fu, J. *et al.* Interface engineering of Ta₃N₅ thin film photoanode for highly efficient photoelectrochemical water splitting. *Nat. Commun.* **13**, 729 (2022).
- 16 Xiao, Y. *et al.* Band structure engineering and defect control of Ta₃N₅ for efficient photoelectrochemical water oxidation. *Nat. Catalysis* **3**, 932-940 (2020).

Reviewer #3

Comments to the Author:

In this study, Bae et al. have investigated the origin of the stability improvement of most commonly used oxide photoanodes (BiVO₄ and Fe₂O₃) via PEI/TiO₂ coating. Coating strategies have previously been suggested to protect the photocorrosion of oxide photoanodes, TiO₂ has been the most common choice of coating overlayer. They have presented a set of experiments and demonstrated that PEI/TiO₂ facilitates hole transfer, despite the surface has low water oxidation activity. Before recommending this paper for publication, the following must be addressed.

Comment #3-1:

100 nm thick TiO₂ overlayers on bare oxide photoanodes act as a hole blocking layer due to the lower band edge, in reversed type II heterojunction configuration. Later, the authors show this in the band diagrams. Therefore, it is not surprising that to get a low photocurrent. It becomes leaky when the overlayer is not uniform, contains pinholes. So, there is many other reasons for “leaky” performance.

Author's response:

(1) “Electronically leaky” amorphous TiO₂

We appreciate the reviewer’s valuable comments. The term “leaky” in the amorphous TiO₂ (a-TiO₂) literature refers to its electronically conductive properties via defect sites within the structure. In general, TiO₂ has a large bandgap (>3 eV) and a deep valence band position, which typically acts as a hole-blocking layer for most semiconductors. However, it is known that a-TiO₂ synthesized by atomic layer deposition (ALD) contains electronic defects formed by C and N atoms from the TDMAT precursor, enabling the anodic conduction of photogenerated holes. For example, Hu et al. (*Science* **2014**, *344*, 1005-1009) demonstrated that an a-TiO₂ protection layer on a Si photoanode resulted in long-term stability exceeding 100 hours. Despite forming an unfavorable heterojunction between the n-p⁺-Si photoanode and a-TiO₂, this configuration exhibited highly effective water oxidation efficiency. In addition, Nunez et al. (*J. Phys. Chem. C* **2019**, *123*, 20116–20129) analyzed the hole conduction mechanism of a-TiO₂ through various solid-state analyses, and they confirmed that the anodic conduction is facilitated by hopping between Ti³⁺ sites and adjacent Ti⁴⁺ sites in the electronic “leaky” a-TiO₂.

In our study, we conducted the same approach by depositing a-TiO₂ with various metal co-catalysts on a metal oxide semiconductor, but photoelectrochemical (PEC) water oxidation was unfeasible. These results suggest the presence of an unknown mechanism in the junction between the photoabsorber and a-TiO₂. Through various characterizations and PEC analyses, we discovered that: 1) the formation of a split Fermi level preventing unfavorable downward band bending and 2) hole-selective contact between the photoabsorber and the defect states of “leaky” a-TiO₂ are crucial for the hole conduction. We believe that these findings provide significant insights into the application of electronically “leaky” a-TiO₂ for low-cost metal oxide PEC devices.

(2) Pinhole test

Despite the ALD technique is known for producing high-quality and uniform films, there is a possibility that pinholes formed during the synthesis of a-TiO₂ could allow the underlying photoabsorber to contact the electrolyte and participate in water oxidation reactions. To investigate this possibility, we carried out an electrochemical pinhole test using a ferricyanide solution (*ACS Appl. Mater. Interfaces* **2017**, *9*, 50, 43614–43622). The a-TiO₂ and the hybrid PEI/TiO₂ were deposited on fluorine-doped tin oxide (FTO) and BiVO₄ substrates, respectively, and the ferricyanide redox reaction was evaluated. In this test, FTO and BiVO₄ showed clear redox peaks, whereas these peaks were absent when 100 nm a-TiO₂ and hybrid PEI/TiO₂ were deposited. These results suggest that the thick (>100nm) hybrid PEI/TiO₂ layer contains very few pinholes, rendering their influence negligible. Therefore, we conclude that the photocurrent observed in photoanodes modified with the hybrid PEI/TiO₂ is not attributable to pinholes or cracks but originates from the water oxidation reaction occurring on the surface of the hybrid PEI/TiO₂.

Revision made (highlighted):

[Supplementary Fig. 2 in the revised manuscript]

Supplementary Fig. 2. SEM images of FTO, FTO/TiO₂, and FTO/PEI/TiO₂. a-c, FTO/TiO₂ and FTO/PEI/TiO₂ were prepared in the same manner as the BiVO₄ photoanode. The PEI layer was deposited on the FTO substrate using spin-coating, followed by the deposition of 100 nm a-TiO₂ on the substrate.

[Supplementary Fig. 3 in the revised manuscript]

Supplementary Fig. 3. Pinhole test using a ferricyanide solution. Pinhole test was carried out using the cyclic voltammetry (scan rate: 50 mV s⁻¹) in the ferricyanide solution including 5 mM K₄[Fe(CN)₆] and 5 mM K₃[Fe(CN)₆] dissolved in 0.5 M KCl solution (pH 7.1). In the test, neither the TiO₂ nor the PEI/TiO₂ overlayers exhibited a redox peak for ferricyanide, which indicates that the FTO (**a**) and the BiVO₄ (**b**) surfaces were fully covered by the overlayers.

[Line 95, page 5 in the revised manuscript]

The conformal and uniform layer of a-TiO₂ was further confirmed by a pinhole test using cyclic voltammetry in a ferricyanide solution (**Supplementary Fig. 2 and 3**).¹⁴

Comment #3-2:

TOF-SIMS analysis shows a significant amount of carbon in BiVO₄/TiO₂ system. The authors acknowledge that the source of carbon is the ALD precursor Ti. TiO₂ has been considered amorphous, but what is the evidence that the overlayer is actually TiO₂, rather than Ti oxide with C mixed matrix?

Author's response:

Typically, TOF-SIMS analysis is considered capable of quantitative analysis only among the same elements due to variations in elemental sensitivity depending on the analysis mode. The use of Cs⁺ ion beams in negative mode, as used in our measurements, enhances the signals of electronegative elements such as chalcogens and halogens, as well as carbon. This explains the significantly higher carbon signal compared to titanium in the BiVO₄/TiO₂ and the BiVO₄/PEI/TiO₂ samples. However, the cross-sectional EDX analysis, which allows for more accurate elemental quantification (**Supplementary Fig. 21 in the revised manuscript**), revealed that both the amorphous TiO₂ and the hybrid PEI/TiO₂ layer had the same elemental composition as TiO₂, with 33% Ti and 66% oxygen atomic percent, respectively. This result indicates that the structure of the amorphous TiO₂ remains unchanged even after forming the hybrid PEI/TiO₂ layer.

(Surf. Interface Anal. 2012, 44, 232–237 / Surf Interface Anal. 2022, 54, 165–173)

Comment #3-3:

In LSV scans under chopped illumination, there is a cathodic peaks at a low potential range once PEI/TiO₂ is present. What would be the origin of this behaviour? It seems like the appearance and disappearance of these cathodic features is correlated with the PEI thickness and therefore the overall photoactivity.

Author's response:

In the LSV measurement, two distinct positive and negative transient peaks can typically be observed under chopped illumination. The positive transient is attributed to the rapid generation and accumulation of photogenerated holes trapped at the semiconductor/electrolyte interface, occurring when these photogenerated carriers do not immediately contribute to the steady-state current. The cathodic peak, or the negative transient, is observed when the light is turned off, and it indicates that the release or recombination of the accumulated holes with electrons. Smaller positive and negative transients suggest better injection of charge into the electrolyte, i.e., higher catalytic activity. However, in our study, we observed a slight increase in the negative transient peak near the onset potential, even though these electrodes delivered high photocurrents under

more positive bias (**Supplementary Fig. 23 in the revised manuscript**). Therefore, there appears to be no correlation between overall photoactivity and the transient peak.

(*Chem. Sci.*, **2014**, *5*, 2964–2973 / *J. Phys. Chem. C* **2012**, *116*, 26707–26720)

Comment #3-4:

In the case of BiVO₄, the back illumination has been performed. What would be the case once front illumination is performed?

Author's response:

The back and frontside illumination depend on the diffusion length of majority/minority carriers and the degree of carrier trapping during charge transport. For BiVO₄, it is well-known that electron transport (even though they are the majority carriers) is significantly hindered by recombination at bulk trap sites near the conduction band (*J. Phys. Chem. Lett.* **2013**, *4*, 2752–2757). Therefore, when BiVO₄ is measured under frontside illumination, most photogenerated electron-hole pairs recombine within the BiVO₄, leading to decreased PEC performance. We addressed this issue by using backside illumination, which allows photogenerated electron-hole pairs to form closer to the FTO substrate, enhancing the chance of electron extraction prior to recombination. In our measurements, BiVO₄, BiVO₄/PEI/TiO₂, and BiVO₄/PEI/TiO₂/CoOOH electrodes exhibited much higher the PEC performance in terms of onset potential and photocurrent under backside illumination, but showed significantly decreased efficiency under frontside illumination.

(*J. Phys. Chem. C* **2012**, *116*, 9398–9404 / *Electrochimica Acta* **2016**, *211* 173–182 / *ACS Energy Lett.* **2023**, *8*, 2177–2184)

Revision made (highlighted):

[REDACTED]

Comment #3-5:

The improved activity has been attributed to the presence of Ti^{3+} species in the PEI matrix. Energy differences in Ti L edge EEL spectra and the noisy VB spectra have been shown as the evidence. The authors must confirm this by comparing Ti 2p XPS spectra. It will be less noisy and Ti^{3+} peak, if exists, would be apparent. Also, the peak shift in Ti L edge EEL spectra is not apparent. The authors must compare at least two spectra from TiO_2 and PEI/ TiO_2 matrix, to justify the peak shift.

Author's response:

To verify the presence of a higher concentration of Ti^{3+} species, we conducted additional EELS and XPS depth profiling measurements. Using Cs-corrected TEM, we performed cross-sectional EELS analyses in three different regions of the $BiVO_4/TiO_2$ and the $BiVO_4/PEI/TiO_2$ samples. The results indicated that the Ti $2p_{1/2}$ and $2p_{3/2}$ peaks of $BiVO_4/PEI/TiO_2$ shifted by an average of -0.4 eV and -0.36 eV, respectively, compared to $BiVO_4/TiO_2$. However, the XPS depth profiling analysis of the Ti 2p peak revealed various Ti oxidation states within the TiO_2 layer, including mixed reduced species of Ti^{4+} , Ti^{3+} and Ti^{2+} in both $BiVO_4/TiO_2$ and $BiVO_4/PEI/TiO_2$. This is attributed to the depth profiling process, which preferentially sputters lighter atoms (i.e., oxygen in the TiO_2), thereby reducing the oxidation states of TiO_2 in situ. Therefore, we tried to fit the Ti $2p_{1/2}$ peak obtained from each electrode surface and found that $BiVO_4/PEI/TiO_2$ showed a larger proportion of Ti^{3+} than that of $BiVO_4/TiO_2$. Based on these results, we demonstrate that the

interfacial PEI layer can be integrated with amorphous TiO₂ during ALD process, thereby forming a hybrid PEI/TiO₂ structure with an increased amount of Ti³⁺. We have replaced Fig. 4 in the revised manuscript with newly obtained EELS data.

Revision made (highlighted):

[Supplementary Fig. 28 in the revised manuscript]

Supplementary Fig. 28. Cross-sectional TEM images of BiVO₄/TiO₂ for EELS analysis. a-c, Cs-corrected TEM measurements of BiVO₄/TiO₂ with the probing path of line scan for EELS.

[Supplementary Fig. 29 in the revised manuscript]

Supplementary Fig. 29. Cross-sectional TEM images of BiVO₄/PEI/TiO₂ for EELS analysis. a-c, Cs-corrected TEM measurements of BiVO₄/PEI/TiO₂ with the probing path of line scan for EELS.

[Supplementary Fig. 30 in the revised manuscript]

Supplementary Fig. 30. The EELS spectra of $\text{BiVO}_4/\text{TiO}_2$. a-c, The EELS spectra of Ti L edge obtained from line scans of three different regions in the $\text{BiVO}_4/\text{TiO}_2$ sample.

[Supplementary Fig. 32 in the revised manuscript]

Supplementary Fig. 32. The EELS spectra of $\text{BiVO}_4/\text{PEI}/\text{TiO}_2$. a-c, The EELS spectra of Ti L

edge obtained from line scans of three different regions in the BiVO₄/PEI/TiO₂ sample.

[Table S2 in the revised manuscript]

Table S2. The average position of the Ti 2p peak of BiVO₄/TiO₂ in the EELS analysis.

BiVO ₄ /TiO ₂	2p _{3/2}	2p _{1/2}
1	459.2	464.5
2	458.8	464.2
3	458.8	464.2
Average	458.93	464.3

[Table S3 in the revised manuscript]

Table S3. The average position of the Ti 2p peak of BiVO₄/PEI/TiO₂ in the EELS analysis.

BiVO ₄ /PEI/TiO ₂	2p _{3/2}	2p _{1/2}
1	458.5	463.8
2	458.7	464.0
3	458.5	463.9
Average	458.57	463.9

[REDACTED]

[Supplementary Fig. 33 in the original manuscript]

Supplementary Fig. 33. High resolution XPS spectra of the Ti 2p peak in BiVO₄/TiO₂ and BiVO₄/PEI/TiO₂. a-b, The Ti 2p peak (a) and Ti 2p_{1/2} peak deconvolution (b) of BiVO₄/TiO₂. c-d, The Ti 2p peak (c) and Ti 2p_{1/2} deconvolution (d) of BiVO₄/PEI/TiO₂. BiVO₄/PEI/TiO₂ showed a higher proportion of Ti³⁺ compared to BiVO₄/TiO₂.

[Fig. 4. in the original manuscript]

[Fig. 4. in the revised manuscript]

[Line 231, page 12 in the revised manuscript]

EELS analysis of $\text{BiVO}_4/\text{TiO}_2$ showed two primary peaks in the Ti L-edge (459.2 and 464.5 eV) and O K-edge spectrum (531.4 and 542.65 eV), and we verified that the position of each peak

remained consistent regardless of the depth (Fig. 4c and Supplementary Fig. 30 and 31).

[Line 239, page 13 in the revised manuscript]

The reduction of Ti was also confirmed through high resolution XPS analysis and Kelvin probe force microscopy (KPFM) measurements for both normal TiO₂ and hybrid PEI/TiO₂. The peak deconvolution of Ti 2p_{1/2} in both electrodes showed a higher proportion of Ti³⁺ in the hybrid PEI/TiO₂ (Supplementary Fig. 33).

Comment #3-6:

Judging by the SEM images presented, there is a significant difference in the morphology of the photoanodes surfaces once coated with PEI/TiO₂. What would be the reason for this? More compact and uniform layers could contribute to the overall activities.

Author's response:

The SEM images reveal that the morphology of BiVO₄/PEI/TiO₂ is more compact and denser compared to BiVO₄/TiO₂ (Supplementary Fig. 1 in the revised manuscript). This difference can be attributed to the formation of TiO₂ on the PEI coated within the pores of BiVO₄. We confirmed that the pores of BiVO₄ were readily filled with the PEI by spin-coating (Supplementary Fig. 4 in the revised manuscript). During the ALD TiO₂ synthesis, the TDMAT precursor is deposited on this PEI layer, resulting in a denser morphology compared to BiVO₄/TiO₂. Nevertheless, as mentioned in the original manuscript, hole transfer from BiVO₄ to the hybrid PEI/TiO₂ is anticipated to occur only through a very thin interfacial PEI layer (<2 nm). Therefore, the pore region is expected to have a negligible effect on the overall activity of the hybrid PEI/TiO₂ in PEC water oxidation.

Comment #3-7:

Can authors exclude the oxidation of PEI under potential/illumination?

Author's response:

To suggest an effective approach for developing a robust hybrid PEI/TiO₂ in future studies, we investigated which type of degradation PEI undergoes under PEC water oxidation conditions. In

our system, PEI is anticipated to exist in three forms: as an interfacial layer, an embedded moiety, and excess PEI on the surface. Among these, it is believed that the excess PEI on the surface readily oxidizes and the cleaved polymer dissolves into the electrolyte. Therefore, we hypothesized that the oxidation of PEI in the $\text{BiVO}_4/\text{PEI}/\text{TiO}_2$ is caused by either the interfacial layer or the embedded PEI within the TiO_2 structure. We first carried out chronoamperometry (CA) measurement of the $\text{BiVO}_4/\text{PEI}/\text{TiO}_2$ without a co-catalyst until the current density reached nearly zero. Subsequently, we measured SEM images of the $\text{BiVO}_4/\text{PEI}/\text{TiO}_2$ to identify any changes in morphology. We supposed that if the embedded PEI moiety decomposed, there could be observable changes in the surface morphology. However, SEM measurements exhibited that the surface morphology of the $\text{BiVO}_4/\text{PEI}/\text{TiO}_2$ remained almost unchanged even after 8 h of CA measurement. In addition, XPS depth profiling measurements to verify the internal composition of the hybrid PEI/TiO_2 revealed no significant difference in nitrogen intensity within the TiO_2 structure. Therefore, we conclude that the main cause is the decomposition of the interfacial PEI layer. We assume that the interfacial PEI layer, including a small amount of internal water will be oxidized during the water oxidation reaction. This result is likely due to slow hole extraction influenced by the absence of co-catalyst, as $\text{BiVO}_4/\text{PEI}/\text{TiO}_2$ demonstrated long-term stability with CoOOH . The low hole extraction leads to changes or degradation of PEI at the interface between BiVO_4 and TiO_2 , hindering selective hole transfer in the $\text{BiVO}_4/\text{PEI}/\text{TiO}_2$. Future studies will focus on incorporating of either more effective co-catalysts or replacing PEI with more stable alternatives that can enhance hole extraction within the interfacial PEI layer.

Revision made (highlighted):

[Supplementary Fig. 43 in the revised manuscript]

Supplementary Fig. 43. Stability test of BiVO₄/PEI/TiO₂ without co-catalyst. The stability of BiVO₄/PEI/TiO₂ without co-catalyst was evaluated by CA measurement at 1.23 V vs. RHE.

[Supplementary Fig. 44 in the original manuscript]

Supplementary Fig. 44. SEM image of BiVO₄/PEI/TiO₂ after stability test. a-b, SEM measurement revealed that a conformal and dense hybrid PEI/TiO₂ remained intact, even after BiVO₄/PEI/TiO₂ reached nearly zero current density following an 8 h stability test.

[Supplementary Fig. 45 in the original manuscript]

Supplementary Fig. 45. XPS depth profiling of BiVO₄/PEI/TiO₂ before and after the stability test. The depth profiling analysis showed identical intensities of N 1s peaks in the hybrid PEI/TiO₂ before and after the stability test. This result indicates that the oxidation of the interfacial PEI layer is the main cause of decreased PEC performance of BiVO₄/PEI/TiO₂ during water oxidation. We assume that the interfacial PEI layer, including a small amount of internal water will be oxidized during the water oxidation reaction.

Reviewer #4

Comments to the Author:

The paper describes a new type of protection layers for photoanodes that is composed of the organic polymer polyethylene imine and ALD processed amorphous TiO₂. The system shows a surprisingly good performance that is certainly be worth communicating. However, the presented discussion of the many data collected in this paper have serious flaws in the discussion/interpretation. Data that actually do not provide a new insight might be omitted. Overall, the manuscript is considerably too long for a communication paper. I recommend reconsideration after a complete revision.

We appreciate the reviewer's valuable comments. Regarding the length of the manuscript, *Nature communication* limits the main text to 5,000 words (not including Abstract, Methods, References and Figure legends), while our manuscript contains only 3,839 words. Therefore, we would kindly request not to modify the length of the main text to ensure a thorough and sufficient explanation of the data.

Comment #4-1:

Lines 97/99 here is a comparison of current density (A/cm²) to current (A). This is not traceable without knowledge of the electrode area. Overall, this should be avoided for better readability. Same problem in Line 196.

Author's response:

Thank you for catching this oversight, we have standardized the unit of current (A) to current density (A cm⁻²).

Revision made (highlighted):

[Line 107, page 6 in the revised manuscript]

Even after the modification with CoOOH co-catalyst (BiVO₄/TiO₂/CoOOH), the photocurrent density only remained in the microampere (μA cm⁻²) range at 1.23 V vs. RHE.

[Line 214, page 11 in the revised manuscript]

The XPS and SEM measurements showed the same trend compared to BiVO₄ (Supplementary

Fig. 25 and 26), and the Fe₂O₃/TiO₂ also exhibited a very low photocurrent density in the microampere scale (0.0018 mA cm⁻² at 1.6 V vs. RHE) similar to that of BiVO₄/TiO₂ (Supplementary Fig. 27a-b).

Comment #4-2:

Line 106, work function of the metal. It seems a bit questionable that under operation condition the surface is formed by a metal. It could also be that the metal is oxide covered under the operating conditions. How did the author establish that the catalyst is in the metallic state?

Author's response:

We also anticipate that the surface of the metals will be oxidized under a positive applied voltage. However, the interface with the TiO₂ (i.e. the embedded part of the metal in the TiO₂) will retain its metallic properties at least for short time. This metal-semiconductor interface is considered crucial for facilitating hole transfer through the “leaky” state in the TiO₂, particularly in Si photoanode. For example, Nunez et al. (*J. Phys. Chem. C* **2019**, *123*, 20116–20129) demonstrated the conductivity across interfaces between metal contacts and amorphous TiO₂ films varies depending on the metal used. Metals with work functions lower than that of the TiO₂ typically provide higher conductivity compared to metals with higher work functions. This is attributed to Ti^{3+/4+} conduction, which uses defect sites near the surface to reduce band bending in the TiO₂ and create either insulating or conducting interfaces. Based on this, we attempted to verify the difference in PEC performance according to the work function by depositing various types of metals on the BiVO₄/TiO₂ (Supplementary Fig. 6 in the revised manuscript). However, the results showed a discrepancy with the previous study, which suggests that there may be so far unknown necessary properties of the interfaces or other underlying issues in hole transfer between the metal oxide electrode and the amorphous TiO₂ protective layer.

Comment #4-3:

Line 112 “Further discussion on the origin of the photocurrent in BiVO₄/PEI/TiO₂ will be discussed in a subsequent section.” [Avoid “overdiscussing” ;-)]

Author's response:

Following the reviewer's suggestion, we have revised the repeated word usage in the main text.

Revision made (highlighted):

[Line 121, page 6 in the revised manuscript]

Further details on the origin of the photocurrent in BiVO₄/PEI/TiO₂ will be discussed in a subsequent section.

Comment #4-4:

The XPS survey spectra are not very informative and might be placed in SI. The Figure S9 should be supplemented by a peak fit. The authors should discuss the origin of an F 1s signal in Figure 2a. In light of the discussion about involvement of PEI in oxidation/degradation, it would be interesting to analyze high resolution spectra of the as prepared electrodes and after prolonged usage.

Author's response:

Following the reviewer's suggestion, we have revised each section in the main text.

(1) The origin of the F 1s signal in BiVO₄

Fluorine-doped tin oxide (FTO), used as a conductive substrate for BiVO₄ in our study, typically shows a F 1s peak around 685 eV in XPS analysis. Based on this, we initially suggested that the peak measured at 682.5 eV of the BiVO₄ corresponded to the F 1s peak from FTO substrate. However, we realized that the F 1s peak was absent in our XPS analysis of bare Fe₂O₃ (**Supplementary Fig. 25 in the revised manuscript**), which suggest that the 682.5 eV peak in BiVO₄ may not originate from FTO. To clarify the origin of this peak, we reviewed previous studies that conducted XPS analysis of BiVO₄ powders (*J. Mater. Chem. A*, **2014**, *2*, 6209-6217 / *Sustainable Energy Fuels*, **2022**, *6*, 1698-1707) and electrodes (*Adv. Mater. Interfaces* **2017**, *4*, 1700540 / *J. Mater. Chem. A*, **2023**, *11*, 24239-24247). These studies also observed a peak around 680 eV, which was identified as the Bi 4p peak. The position of Bi 4p peak reported in the reference paper showed better correspondence with that of observed in our BiVO₄ electrode compared to the F 1s. Therefore, based on these previous reports, we revised the identification of the 682.5 eV peak in BiVO₄ to Bi 4p. We appreciate the reviewer's valuable comments to clarify this observation.

(2) The XPS data in the main text

The XPS data (**Fig. 2a**) presented in the main text demonstrate that the TiO₂ protection layer on the surface of porous BiVO₄ electrode is highly uniform and conformal. The disappearance of Bi and V peaks in BiVO₄/TiO₂ and BiVO₄/PEI/TiO₂ indicates that the exposed BiVO₄ surface is completely covered by the TiO₂ overlayers. These results also confirm that water oxidation reaction cannot occur on BiVO₄, but exclusively on the surface of the TiO₂ and hybrid PEI/TiO₂ layer in the BiVO₄/TiO₂ and BiVO₄/PEI/TiO₂ electrodes. Therefore, we propose including this figure in the main text for better clarity and emphasis.

(3) Peak fitting analysis of the carbon in the XPS data

Following the reviewer's suggestion, we carried out deconvolution for each carbon peak observed in BiVO₄/TiO₂ and BiVO₄/PEI/TiO₂. The XPS data of BiVO₄/TiO₂ revealed two distinct peaks at 286.1 eV and 288.2 eV, corresponding to C-O and O-C=O, respectively. However, we observed that these peaks were negatively shifted to 285.6 eV and 287.4 eV in the BiVO₄/PEI/TiO₂. The shifted peaks were identified as C-NH₂ and C-NHR, which indicate that carbon is bonded to nitrogen functional groups. This result suggests that PEI is incorporated into the TiO₂ layer during the ALD process.

(4) High resolution spectra and depth profiling analysis

Following the reviewer's suggestion, we conducted high resolution XPS measurements of the Ti 2p peaks before and after the water oxidation reaction. Before the reaction, we clearly observed the Ti 2p_{1/2} and Ti 2p_{3/2} peaks in both BiVO₄/TiO₂ and BiVO₄/PEI/TiO₂, with a higher proportion of Ti³⁺ in the Ti 2p_{1/2} peak of BiVO₄/PEI/TiO₂. Both Ti 2p peaks remained even after 120 h of stability test, and the proportion of Ti³⁺ in the Ti 2p_{1/2} peak was unchanged. In addition, we conducted experiments to evaluate the oxidation and degradation of PEI using the BiVO₄/PEI/TiO₂ electrode without a co-catalyst. Chronoamperometry (CA) measurements of the BiVO₄/PEI/TiO₂ electrode were performed over 8 h, after which the photocurrent approached nearly zero. We assumed that this result was due to the degradation of either the interfacial PEI layer or the PEI embedded within TiO₂. To confirm this hypothesis, we carried out SEM and XPS depth profiling analyses on the electrode (**Comment# 3-7**). SEM images revealed negligible changes in the

electrode morphology, and XPS depth profiling analysis showed identical nitrogen peak intensities before and after the reaction. Based on these results, we propose that the degradation of PEI in the $\text{BiVO}_4/\text{PEI}/\text{TiO}_2$ system is attributable to the oxidation of the interfacial PEI layer by accumulated photogenerated holes. Consequently, our future studies should focus on developing a durable interlayer that can more effectively extract accumulated holes from the hybrid PEI/ TiO_2 system.

Revision made (highlighted):

[Fig. 2a. in the original manuscript]

[Fig. 2a. in the revised manuscript]

[Line 154, page 8 in the revised manuscript]

Next, we determined the elemental composition of the photoanodes before and after deposition of TiO_2 and PEI layer by X-ray photoelectron spectroscopy (XPS) analysis (Fig. 2a). The peaks of

Bi (4f, 4d, and 4p), V (2p), and O (1s) were observed in the bare BiVO₄.

[Supplementary Fig. 19 in the revised manuscript]

Supplementary Fig. 19. Peak deconvolution of C 1s spectra in BiVO₄/TiO₂ and BiVO₄/PEI/TiO₂. a-b, BiVO₄/PEI/TiO₂ showed negatively shifted peaks of C-O and O-C=O from 286.1 eV and 288.2 eV to 285.6 eV and 287.4 eV. This result indicates the presence of carbon bonded to amine-based functional groups. Specifically, the shifted peaks at 285.6 eV and 287.4 eV correspond to C-NH₂ and C-NHR functional groups, respectively.

[Line 160, page 8 in the revised manuscript]

BiVO₄/PEI/TiO₂ also indicated a similar trend with BiVO₄/TiO₂ but showed negatively shifted C peaks at 285.6 eV and 287.4 eV, corresponding to C-NH₂ and C-NHR, respectively, along with stronger N peaks (Supplementary Fig. 18 and 19).^{24,25}

[Supplementary Fig. 33 in the revised manuscript]

Supplementary Fig. 33. High resolution XPS spectra of the Ti 2p peak in $\text{BiVO}_4/\text{TiO}_2$ and $\text{BiVO}_4/\text{PEI}/\text{TiO}_2$. a-b, The Ti 2p peak (a) and Ti 2p_{1/2} peak deconvolution (b) of $\text{BiVO}_4/\text{TiO}_2$. c-d, The Ti 2p peak (c) and Ti 2p_{1/2} deconvolution (d) of $\text{BiVO}_4/\text{PEI}/\text{TiO}_2$. $\text{BiVO}_4/\text{PEI}/\text{TiO}_2$ showed a higher proportion of Ti^{3+} compared to $\text{BiVO}_4/\text{TiO}_2$.

[Supplementary Fig. 16 in the revised manuscript]

Supplementary Fig. 16. XPS analysis of BiVO₄/PEI/TiO₂/CoOOH after the stability test. a-b, High resolution XPS spectra of BiVO₄/PEI/TiO₂/CoOOH after 120 h stability test. XPS analysis showed a clear Ti 2p peak including Ti³⁺ state in BiVO₄/PEI/TiO₂/CoOOH.

[Supplementary Fig. 44 in the revised manuscript]

Supplementary Fig. 44. SEM image of BiVO₄/PEI/TiO₂ after stability test. a-b, SEM measurement revealed that a conformal and dense hybrid PEI/TiO₂ remained intact, even after BiVO₄/PEI/TiO₂ reached nearly zero current density following an 8 h stability test.

[Supplementary Fig. 45 in the revised manuscript]

Supplementary Fig. 45. XPS depth profiling of BiVO₄/PEI/TiO₂ before and after the stability test. a-b, The depth profiling analysis showed identical intensities of N 1s peaks in the hybrid PEI/TiO₂ before and after the stability test. This result indicates that the oxidation of the interfacial PEI layer is the main cause of decreased PEC performance of BiVO₄/PEI/TiO₂ during water oxidation. We assume that the interfacial PEI layer, including a small amount of internal water will be oxidized during the water oxidation reaction.

Comment #4-5:

Line 183, “These 183 results suggest that variations in the measured photocurrent mainly stem from the amount of 184 embedded C and N within the hybrid PEI/TiO₂ layer.” This sentence should be deleted. A correlation does not mean causality. PEI is not a crystalline compound, in which elemental composition determine crystal structure and electronic properties. It is a molecular compound whose properties depend mainly on the connectivity of atoms. In addition, SIMS (and XPS) cannot determine the content of H, which is contained in PEI at a high mole fraction. The authors may take another organic material with similar atomic N:C ratio as PEI and

try the effect. Most likely such a compound would not at all be functional.

Author's response:

We agree that correlation does not imply causality and that the elemental composition of PEI does not determine crystal structure and electronic properties. What we intended to say is that achieving both the widened defect bands and the split Fermi level between a photoabsorber and the TiO₂ requires a “sufficient amount of incorporated PEI”. For example, XPS and EELS analyses indicated that incorporating PEI into the TiO₂ resulted in the formation of reduced Ti³⁺ species, which widened the defect bands and thereby facilitated hole transfer. We agree with the reviewer's opinion that these results are not attributable to “the amounts of carbon and nitrogen”, but stem from “the connectivity of atoms” (likely the bonding between N and Ti). Following the reviewer's suggestion, we have removed these from the sentence and restructured it entirely.

Revision made (highlighted):

[Line 183, page 10 in the original manuscript]

These results suggest that variations in the measured photocurrent mainly stem from the amount of embedded C and N within the hybrid PEI/TiO₂ layer.

[Line 204, page 10 in the revised manuscript]

These results imply that the amount of embedded PEI may influence the properties of the hybrid PEI/TiO₂, thereby contributing to the PEC performance.

[Line 203, page 11 in the original manuscript]

It was assumed that the presence of C and N in hybrid PEI/TiO₂ could influence the intrinsic properties of amorphous TiO₂, thereby contributing to facilitate hole transfer.

[Line 224, page 12 in the revised manuscript]

It was assumed that the presence of PEI moiety in hybrid PEI/TiO₂ could influence the intrinsic properties of amorphous TiO₂.

Comment #4-6:

Have the authors evaluated the possibility that the PEI layer modulates the pH at the interface that could have a very pronounced effect on the photocurrent? Under operating condition, the local pH will deviate from the bulk pH. PEI may support or counteract this effect.

Author's response:

In our previous research, we found that the protonation of abundant amine group in the PEI on the photoelectrode surface could improve water oxidation efficiency through local pH changes (*Adv. Mater. Interfaces* **2023**, *10*, 2202101). The PEI layer was deposited on the electrode in the form of a hydrogel cross-linked with glutaraldehyde, contributing to relatively high durability against degradation during the water oxidation reaction. However, in the case of hybrid PEI/TiO₂, it is expected that only a very small amount of PEI is present on the electrode surface. Additionally, since the PEI could be directly exposed to the electrolyte and easily oxidized, its effect on the local pH is unlikely to be sustained for long. To confirm this, we first deposited the PEI directly on the BiVO₄ electrode by spin-coating and conducted LSV and CA measurement. The LSV curve showed a peak for PEI oxidation around 0.7 V vs. RHE, and the CA measurement indicated a rapid decrease in the photocurrent within 5 h. Furthermore, we observed even a slight increase in terms of the photocurrent after removing the residual PEI on the BiVO₄/PEI/TiO₂ surface through UV/Ozone treatment. Based on these results, we deduce that the small amount of residual PEI on the hybrid PEI/TiO₂ surface has a negligible effect on local pH changes at the electrode surface.

Revision made (highlighted):

[REDACTED]

[REDACTED]

Comment #4-7:

P. 15, EIS; it remains completely unclear what was actually done and how the authors can arrive at their conclusions. Which potential was externally applied while making the EIS measurement? If no potential was applied, what was the open circuit potential that is established spontaneously? Which electrocatalytic reactions run at the DC potential? If no reaction is running, how can the impedance data reveal anything about catalytic activity?

Author's response:

We thank the reviewer for pointing out this oversight. In the EIS analysis, we applied a DC potential from 0.3 V to 1.4 V vs. RHE with 0.1 V intervals. The resistance and capacitance values (**Supplementary Fig. 40 in the revised manuscript**) were obtained by fitting the EIS data measured at each DC potential step (**Supplementary Fig. 41 in the revised manuscript**). All measurements were conducted under illumination, and the LSV curve included in the resistance graph confirmed that PEC water oxidation occurred (and was recorded directly after the EIS measurement). Through the EIS analysis, we attempted to analyze the reason of the PEI degradation in the absence of a cocatalyst. In the revised manuscript, we incorporated new conclusions based on the newly obtained data to provide a more comprehensive explanation.

Revision made (highlighted):

[**Line 290, page 15** in the revised manuscript]

In the DWE and KPFM analyses, although we confirmed that selective hole transfer is facilitated by the large electron barrier height of the interfacial PEI layer, we were concerned the oxidation of PEI by photogenerated holes contributed by the small hole barrier height. For instance, a sulfite oxidation measurement revealed that BiVO₄/PEI/TiO₂ shows a transient in the photocurrent even for the expected fast reaction kinetics of the sacrificial electron donor, while it disappeared in the presence of a CoOOH co-catalyst (**Supplementary Fig. 39**). This result implies either low catalytic activity or slow hole extraction of the hybrid PEI/TiO₂, leading to the oxidation of the interfacial PEI layer during water oxidation, as we mentioned earlier. Therefore, we first conducted electrochemical impedance spectroscopy (EIS) for a comparative analysis between the BiVO₄/PEI/TiO₂ and BiVO₄/PEI/TiO₂/CoOOH electrodes to evaluate the catalytic activity of the hybrid PEI/TiO₂ in water oxidation (**Supplementary Fig. 40**). The Nyquist plot was fitted by employing a series 3RC-equivalent circuit to determine the resistance and capacitance values in each frequency domain (**Supplementary Fig. 41 and 42**).^{19,40} BiVO₄/PEI/TiO₂ and BiVO₄/PEI/TiO₂/CoOOH exhibited nearly identical resistance and capacitance values in the high (HF) and medium-frequencies (MF), but we observed a significant difference of resistance in the low-frequency (LF) related to the charge transfer resistance into the electrolyte and therefore the water oxidation kinetics. While BiVO₄/PEI/TiO₂ maintained a higher resistance even at high

applied potential, the resistance of $\text{BiVO}_4/\text{PEI}/\text{TiO}_2/\text{CoOOH}$ decreased steeply after the onset potential (0.28 V vs. RHE) (**Supplementary Fig. 40a-b**). This result shows that $\text{BiVO}_4/\text{PEI}/\text{TiO}_2$ has relatively low catalytic activity, emphasizing the need for an effective cocatalyst on the hybrid PEI/TiO_2 layer to achieve stable solar water oxidation. Our findings were confirmed in the stability test of $\text{BiVO}_4/\text{PEI}/\text{TiO}_2$ without co-catalyst. In the CA measurement, we observed that the current density of $\text{BiVO}_4/\text{PEI}/\text{TiO}_2$ reached nearly zero after 8 h (**Supplementary Fig. 43**), despite the surface morphology of the electrode remaining intact (**Supplementary Fig. 44**). In addition, XPS depth profiling measurements of the hybrid PEI/TiO_2 revealed no significant difference in nitrogen intensity within the TiO_2 structure (**Supplementary Fig. 45**). This result points to the fact that the primary cause is the decomposition of the interfacial PEI layer. The decomposition is likely due to slow hole extraction influenced by the absence of co-catalyst, as $\text{BiVO}_4/\text{PEI}/\text{TiO}_2$ demonstrated long-term stability with CoOOH . The slow hole extraction leads to changes or degradation of PEI at the interface between BiVO_4 and TiO_2 , hindering selective hole transfer in the $\text{BiVO}_4/\text{PEI}/\text{TiO}_2$. Consequently, future studies should focus on incorporating more effective co-catalysts or replacing PEI with more stable alternatives to further enhance the hole extraction within the interfacial PEI layer.

Response to Reviewers' Comments

(NCOMMS-24-16729-T)

We thank the reviewers again for their comments, which have helped to improve the quality and clarity of this manuscript. Below is our point-by-point response to the Reviewers' comments. Our response to the comments and any changes made to the original manuscript were highlighted in blue and green, respectively.

Reviewer #4

Comments to the Author:

The authors have amended their manuscript by many important data and considerably improved the quality of the discussion formally and scientifically. While this applies to most data, the reviewer is still not satisfied with the interpretation of the XPS data.

Supplementary Figure 16 and Supplementary Figure 33b: It is clear that there is a Ti 2p signal confirming the statement on p7 of the manuscript. For this statement, Supplementary Figure 16a would fully suffice. However, Supplementary Figure 16b contains a fit of the Ti 2p_{1/2} component. This fit interprets the broad peak as indication of two Ti binding states. However, the Ti 2p_{3/2} and Ti 2p_{1/2} components must have an area ratio of 2:1. As the peak height for the Ti 2p_{1/2} component is smaller, the width necessarily must be broader without automatically implying different binding states. Different binding states should also be evident from the Ti 2p_{3/2} component. The reviewer admits that the full interpretation of Ti 2p spectra can be challenging due to suboxides and other satellites. The handling of the baseline is also not trivial. However, when attempting a fitting, it must be based on a physical basis and this should be spelled out (probably in the SI). The presented fitting implies a different spin-orbit splitting depending on the chemical state of the oxide. All this comes without literature reference or other justification, which the reviewer finds unacceptable.

While the data itself in Supplementary Figure 16a support the statement on p7, this is not the case for Supplementary Figure 33b with respect to the statement on p 13. Here all the deficits listed above directly impact on the validity of the conclusion. For this reason, the paper cannot be recommended in the present state.

As a further point, the reviewer recommends caution with destructive XPS depth profiling (Supplementary Figure 45). The chemical states of the compounds is usually *not* preserved and

such conclusion should not be made or implied. A clarifying statement in this regards would help to avoid ambiguities with the readers.

Author's response:

We appreciate the reviewers' constructive suggestions for the XPS analysis, which required corrections and modifications. We would like to address the concerns raised about the data fitting and provide a justification for our interpretation.

(1) Supplementary Fig. 16

In response to the reviewer's suggestion, we have excluded **Supplementary Fig. 16b** from the revised manuscript, which was originally used to confirm the durability of the hybrid PEI/TiO₂ layer after the long-term stability test.

Revision made (highlighted):

[**Supplementary Fig. 16** in the revised manuscript]

Supplementary Fig. 16. XPS analysis of BiVO₄/PEI/TiO₂/CoOOH after the stability test. High resolution XPS spectra of BiVO₄/PEI/TiO₂/CoOOH after 120 h stability test. XPS analysis showed clear Ti 2p peaks in BiVO₄/PEI/TiO₂/CoOOH.

(2) Peak deconvolution of Ti 2p peaks (Supplementary Fig. 33)

We agree with the reviewer's comment. Theoretically, the Ti 2p_{3/2} and 2p_{1/2} peaks should exhibit a 2:1 ratio due to spin-orbital splitting. However, the normalized Ti 2p peaks in BiVO₄/TiO₂ and BiVO₄/PEI/TiO₂ revealed that the height of the 2p_{1/2} peak in BiVO₄/PEI/TiO₂ was relatively lower despite having the same width (the peak area of each Ti 2p_{3/2} was nearly identical, see **Figure R1** below). This trend was further observed by additional XPS measurements, and we considered that one of the samples may not maintain the expected 2:1 ratio in the Ti peaks. Therefore, we had tried to use a flexible basis spline background instead of a Shirley background, which includes the peak area factor, for the partial XPS fitting of Ti 2p_{1/2} peak in the original manuscript (*J. Vac. Sci. Technol. A*, **2020**, *38*, 063203).

However, we have taken the reviewer's suggestion into account and conducted the full interpretation of Ti 2p spectra to achieve a more precise XPS analysis in the revised manuscript. We applied a smoothing process of the original raw XPS data to reduce noise, followed by the XPS fitting with Shirley background correction and appropriate restraints on the fitting (as detailed in the caption of **Supplementary Fig. 33**) in accordance with previous studies (*Surf. Interface Anal.*, **1998**, *26*, 549-564 / *Appl. Surf. Sci.*, **2010**, *257*, 887-898 / *J. Mater. Res.*, **2017**, *32*, 1563-1572). The revised XPS data still indicates a relatively higher Ti³⁺ ratio in BiVO₄/PEI/TiO₂ (**Supplementary Fig. 33 in the revised manuscript**). It is often challenging to precisely measure the distinct peak of partially reduced Ti³⁺, as XPS analysis detects the average peak intensity of each species in the samples (*Sci. Rep.*, **2018**, *8*, 9255). Consequently, when the Ti³⁺ species are present in low quantities, they are typically analyzed through peak deconvolution in the low binding energy region of each Ti peak. This is well-established in previous studies on reduced TiO₂, such as blue (*J. Mater. Chem. A*, **2021**, *9*, 4822-4830) or black (*J. Mater. Chem. A*, **2023**, *11*, 25429-25440) TiO₂, and our findings are also consistent with the previous results. Moreover, the presence of Ti³⁺ species in the hybrid PEI/TiO₂ was also observed through KPFM (**Supplementary Fig. 34 in the original manuscript**), EELS (**Fig. 4 in the original manuscript**), and absorbance (**Supplementary Fig. 36 in the original manuscript**) measurements, and these characterizations further support the results of XPS deconvolution.

[REDACTED]

[Supplementary Fig. 33 in the revised manuscript]

Supplementary Fig. 33. High resolution XPS spectra of the Ti 2p peak in BiVO₄/TiO₂ and BiVO₄/PEI/TiO₂. a-b, The Ti 2p peak deconvolution of BiVO₄/TiO₂ (a) and BiVO₄/PEI/TiO₂ (b). BiVO₄/PEI/TiO₂ showed a slightly higher proportion of Ti³⁺ compared to BiVO₄/TiO₂. To confirm the peak proportion, peak fitting was carried out under the following conditions: Smoothing of the

raw data, Shirley background subtraction with an average width of 1, an area constraint ratio of 1:2 between $2p_{1/2}$ and $2p_{3/2}$, and position constraints of 5.7 for the Ti^{4+} and Ti^{3+} peaks in TiO_2 .

[Line 240, page 13 in the revised manuscript]

The peak deconvolution of $Ti\ 2p_{1/2}$ and $2p_{3/2}$ in both electrodes showed a slightly higher proportion of Ti^{3+} in the hybrid PEI/ TiO_2 (Supplementary Fig. 33).^{30,31}

(3) Supplementary Fig. 45 (The chemical states of the nitrogen in the XPS depth profiling)

We appreciate the reviewer's valuable comments. It is known that chemical states of the elements would likely to be changed during XPS depth profiling due to primary ion implantation, atomic mixing, preferential sputtering, and/or bond breaking associated with decomposition (*Appl. Surf. Sci.*, **2001**, 179, 307-315). Following the reviewer's suggestion, we have replaced "nitrogen" to "nitrogen species" and provided additional explanations regarding the changes in the chemical states of nitrogen to avoid any potential ambiguities for the readers.

[Line 313, page 16 in the revised manuscript]

In addition, XPS depth profiling measurements of the hybrid PEI/ TiO_2 revealed the presence of nitrogen species within the TiO_2 structure (Supplementary Fig. 45).

[Supplementary Fig. 45 in the revised manuscript]

Supplementary Fig. 45. XPS depth profiling of BiVO₄/PEI/TiO₂ before and after the stability test. a-b, Nitrogen species are observed in both samples. We cannot be sure about the identity of the nitrogen species within the film since the depth profiling process often alters chemical states. Nevertheless, the depth profiling analysis showed identical intensities of nitrogen species in the hybrid PEI/TiO₂ before and after the stability test. This result suggests that the oxidation of the interfacial PEI layer is the main cause of decreased PEC performance of BiVO₄/PEI/TiO₂ during water oxidation. We assume that the interfacial PEI layer, including a small amount of internal water will be oxidized during the water oxidation reaction.